# Monsoon Effects on Chlorophyll-a, Sea Surface Temperature, and Ekman Dynamics Variability along the Southern Coast of Lesser Sunda Islands and Its Relation to ENSO and IOD Based on Satellite Observations

**Febryanto Simanjuntak** [1,2] and **Tang-Huang Lin** [2,3,*]

1 Malikul Saleh Meteorological Station, The Agency for Meteorology, Climatology, and Geophysics of the Republic of Indonesia (BMKG), Jl. Bandara Malikussaleh, Muara Batu, Aceh Utara 24355, Indonesia; 109022602@cc.ncu.edu.tw

2 Center for Space and Remote Sensing Research, National Central University, Taoyuan City 32001, Taiwan

3 Center for Astronautical Physics and Engineering, National Central University, Taoyuan City 32001, Taiwan

* Correspondence: thlin@csrsr.ncu.edu.tw; Tel.: +886-03-425-7232

**Abstract:** The Asian–Australian Monsoon (AAM), the El Nino-Southern Oscillation (ENSO), and the Indian Ocean Dipole (IOD) have been known to induce variability in ocean surface characteristics along the southern coast of Lesser Sunda Island (LSI). However, previous studies used low spatial resolution data and little Ekman dynamics analysis. This study aims to investigate the direct influence of AAM winds on ocean surface conditions and to determine how ENSO and IOD affect the ocean surface and depth with higher spatial resolution data. In addition, the variability in Ekman dynamics is also described along with the inconsistent relationship between wind and sea surface temperature (SST) in four different areas. The results indicate that persistent southeasterly winds are likely to induce low SST and chlorophyll-a blooms. Based on the interannual variability, the positive chlorophyll-a (up to 0.5 mg m$^{-3}$) and negative SST (reaching $-1.5\,^\circ$C) anomalies observed in the El Nino of 2015 coincide with +IOD, which also corroborates positive wind stress and Ekman Mass Transport (EMT) anomalies. In contrast, the La Nina of 2010 coincides with -IOD, and positive SST and negative chlorophyll-a anomalies (more than 1.5 $^\circ$C and $-0.5$ mg m$^{-3}$ respectively) were observed. Furthermore, we found that southern coast of Java and Bali Island have a different generated mechanism that controls SST variability.

**Keywords:** upwelling; Ekman dynamics; wind; ENSO; IOD

## 1. Introduction

Upwelling is the process of lifting water mass from a subsurface layer to a higher layer in a water body. This elevated water mass brings cold water to the surface of the sea, in addition to nutrient-rich water that promotes phytoplankton development. As a result, the upwelling region is frequently identified by a higher sea surface chlorophyll-a (chl-a) concentration and a lower sea surface temperature (SST). This, in turn, promotes phytoplankton blooms [1–4] and increases fishery productivity [5–10]. Identifying the spatio-temporal variability of upwelling is crucial, particularly for fishery resources near coastal areas, because it may be utilized as a forecasting indicator, which is highly useful in developing effective fishery management plans [11].

The southern coast of Lesser Sunda Island (LSI) has been noted as one of the most biodiverse regions in the world (referred to as the crosshatch area). Savu Sea is located in this area, bound by Rote Island and Timor bound in the east (marked by R and T in Figure 1, respectively), Savu Island to the south (marked by Su in Figure 1), Sumba Island to the west (marked by S in Figure 1), and Flores and Alor Island to the north and northeast (marked by F and A in Figure 1, respectively). The Savu Sea is part of a coral reef that contains

the world's most remarkable diversity of corals and reef fish species [12,13]. Furthermore, the Savu Sea is a main site for yellow-fin tuna, which is among the most valuable fish resources in the fishing sector [14]. The Savu Sea was designated as a Marine National Park by the Indonesia Ministry of Marine Affairs and Fisheries in 2009, due to its potential marine resources. The fisheries in this area can produce up to 156,000 tons per year [15,16]. However, the maximum level of exploitation of fish has yet to be attained [17–19].

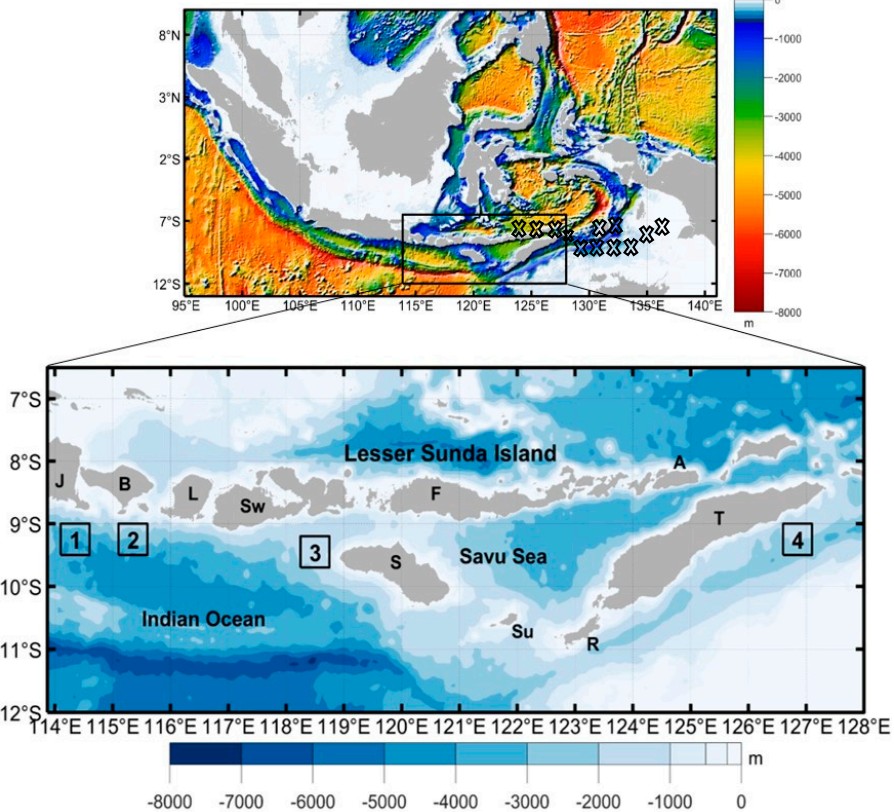

**Figure 1.** Map of the study site at the southern coast of Lesser Sunda Island. The names of islands (marked by alphabet letters) are Java (J), Bali (B), Lombok (L), Sumbawa (Sw), Flore (F), Alor (A), Sumba (S), Savu (Su), Rote (R), and Timor (T). The background color is the bathymetry depth. Black boxes are the specific study area for analyzing the correlation between sea surface temperature (SST) and wind.

The atmospheric dynamics in Indonesia, precisely in LSI, are significantly influenced by the Asian–Australian monsoon system, which consists of the southeast monsoon season from June to August and the northwest monsoon season from December to February [20–25]. The Asian monsoon is characterized by northwesterly winds blowing from Asia to Australia that bring humid air, since it passes through the South China Sea and causes a rainy season in most Indonesian areas. In contrast, the Australian monsoon is characterized by southeasterly winds blowing from Australia to Asia that bring dry air, since it passes through a massive desert area over the northern part of Australia and causes a dry season in most Indonesia areas [26–29]. The variation in the Asian–Australian Monsoon (AAM) system regulates the oceanographic conditions in the Indonesia seas, particularly the southern coast of LSI [11]. Through the mechanism of offshore Ekman Mass Transport, this AAM monsoon system induces coastal upwelling [30]. The upwelling brings cold and rich-nutrient water to the ocean surface, causing phytoplankton blooms, which are fed by higher trophic-level organisms such as fish [31,32]. As a result, fish abundance is usually found during the southeast monsoon season [33].

Furthermore, the upwelling along the southern coast of LSI is caused not only by the southeast monsoon season but also by the El Nino-Southern Oscillation (ENSO) and

the Indian Ocean Dipole (IOD) [34,35]. ENSO is a dominant mode of interannual climate variability that originates from air–sea interactions in the tropical Pacific [36,37]. Through ENSO-related atmospheric teleconnections, ENSO may have an enormous impact on global climate [38,39]. The impact of ENSO on the oceanographic conditions within the Indonesia Seas has been investigated [11,30,40–43]. During El Nino, the SST is cooler than average, and the chl-a concentration is higher than average. In contrast, the SST is warmer, and the chl-a concentration is lower than average during La Nina. This can be understood because the thermocline layer is shallower during El Nino and deepens during La Nina [30]. Furthermore, different types of positive IOD occurrences cause diverse ocean circulation responses, resulting in different patterns in chl-a anomalies in the Indian Ocean [44]. During a positive IOD event, the SST is cooler than average, and the chl-a concentration is higher than average. In contrast, the SST is warmer, and the chl-a concentration is lower than average during negative IOD events.

In terms of atmospheric dynamics, the physical factors that regulate the oceanographic conditions along the southern coast of LSI have been discussed [41]. However, this previous study used low spatial resolution data and homogeneous drag coefficients for weak and strong winds. Therefore, in our analysis, we utilized the most recent sea surface wind product from the Copernicus Marine Service, which has a higher spatial resolution and distinguishes the drag coefficient for weak and strong winds. In addition, to the best of our knowledge, Ekman dynamics, which include Ekman Mass Transport (EMT) and Ekman Pumping Velocity (EPV), were not discussed in detail in the previous study, despite the fact that Ekman dynamics is well known to influence coastal upwelling along the southern coast of LSI [11]. The influence of Ekman dynamics induced by southeasterly winds in causing upwelling variability is also described in this study.

The structure of this paper is as follows. The satellite observations and reanalysis products, and the methods used, are described in Section 2. The results are presented in Section 3, including (1) the seasonal variability of chl-a, SST, EMT, and EPV; (2) the four distinctive areas that have different responses between SST and wind; and (3) the interannual variability along the south coast of the LSI. Sections 4 and 5 highlight the discussion and the conclusions.

## 2. Materials and Methods

### 2.1. Data

The monthly mean values of chl-a and SST data were acquired from Aqua Moderate Resolution Imaging Spectroradiometer (MODIS) at a $0.4° \times 0.4°$ spatial resolution from 2007 to 2020, and can be downloaded from https://giovanni.gsfc.nasa.gov/giovanni/ (accessed on 25 February 2021). Mainly, we selected the MODIS SST obtained from 11 μm, since the daytime and night-time observations are available in this product. The chl-a and SST algorithms were discussed in [45,46], respectively.

The sea surface wind data were obtained from a Met-Op ASCAT satellite with a $0.125° \times 0.125°$ spatial resolution in the form of descending and ascending on a daily basis provided by the Copernicus Marine Service, which can be obtained from https://resources. marine.copernicus.eu/product-dowload/WIND_GLO_WIND_L3_REP_OBSERVATIONS_ 012_005 (accessed on 25 February 2021). This wind product is produced using a microwave scatterometer sensor that can estimate wind speed and direction at a 10 m height based on surface roughness. Furthermore, this product has been examined and evaluated in coastal regions and open seas and has a high level of accuracy [47].

We also used in situ observations of the vertical temperature profile from any available Argo float [48] and the World Ocean Database (WOD) 2018, which can be obtained from https://www.ncei.noaa.gov/products/world-ocean-database (accessed on 25 February 2021) [49]. However, few measurements of the LSI area can be found in these sources. Therefore, we used mixed layer depth data, and the temperature and salinity data were obtained from the reanalysis data, namely, GLOBAL_REANALYSIS_PHY_001_030, which was processed with the NEMO (Nucleus for European Modelling of the Ocean) platform by the Copernicus Marine Service,

found at https://resources.marine.copernicus.eu/product-download/GLOBAL_REANALYSIS_PHY_001_030 (accessed on 25 February 2021). This product has a 0.083° × 0.083° spatial resolution and is at 50 standard levels [50]. Additionally, we calculated the mixed layer depth from the Argo float using a temperature difference of 0.2 °C from the near-surface (10 m) to a given depth [51].

All of the daily data was averaged into monthly mean values before they were used for monthly climatology calculations following this equation [42]:

$$\bar{X}(a,b) = \frac{1}{n} \sum_{j=1}^{n} x_j(a,b,t) \tag{1}$$

where $\bar{X}(a,b)$ denotes the monthly mean or climatological monthly mean at a specific area (a,b), $x_j(a,b,t)$ is the jth value of the data over specific area (a,b) and time t, and n denotes the quantity of daily data in a month or the quantity of monthly data in a climatology timespan (i.e., January 2020 = 31 daily data).

### 2.2. Methods

The sea surface wind data were converted into wind stress (τ) as follows:

$$\tau = \rho_a C_d U_{10}^2 \tag{2}$$

The drag coefficient ($C_d$) value is determined as follows [52]:

$$1000 C_d = 1.29 \text{ for } 0 \text{ m s}^{-1} < U_{10} < 7.5 \text{ m s}^{-1} \tag{3}$$

$$1000 C_d = 0.8 + 0.0065 U_{10} \text{ for } 7.5 \text{ m s}^{-1} < U_{10} < 50 \text{ m s}^{-1} \tag{4}$$

where $\rho_a$ is the air density (1.25 kg m$^{-3}$), and $U_{10}$ is the wind speed at a 10 m height. *EMT* and *EPV* are calculated as follows [53]:

$$EMT = -\frac{\tau}{\rho_w f} \tag{5}$$

$$curl = \frac{\partial \tau_y}{\partial x} - \frac{\partial \tau_x}{\partial y} \tag{6}$$

$$EPV = -\frac{curl}{\rho_w f} \tag{7}$$

where $\rho_w$ is the density of the water (1025 kg m$^{-3}$), and f denotes the Coriolis parameter [54]. *EPV* and *EMT* are measured in m s$^{-1}$ and m$^2$ s$^{-1}$, respectively.

In this study, we also conducted a Pearson linear correlation analysis of monthly data during the southeast monsoon season from 2007 to 2020 to define the role of EMT and EPV in regulating upwelling along the southern coast of LSI (the narrow band south of the entire island chain). In addition, to examine the combination role of EMT and EPV in determining SST variability, we also calculated a multiple correlation coefficient by using EPV and EMT as independent variables and SST as a dependent variable. Furthermore, we also calculated the seawater density based on the vertical profile of temperature and salinity by following the equation given by UNESCO [55] to describe upwelling in detail.

In order to investigate the influence of ENSO and IOD on the ocean conditions in the LSI, we utilized the Oceanic Nino Index (ONI) and Dipole Mode Index (DMI), respectively Figure 2. The ONI data can be downloaded at https://origin.cpc.ncep.noaa.gov/products/analysis_monitoring/ensostuff/ONI_v5.php (accessed on 25 February 2021). ONI can be defined by the SST anomaly in the Nino 3.4 region (5°N–5°S, 170°W–120°W) based on centered 30-year base periods updated every 5 years. El Nino and La Nina events can be distinguished based on an ONI threshold of ±0.5 °C. Additionally, the weekly DMI can be found at https://stateoftheocean.osmc.noaa.gov/sur/ind/dmi.php (accessed on 25 February 2021).

DMI can be defined by the anomaly value of SST gradient between the western tropical Indian Ocean (10°S–10°N and 50°E–70°E) and the southeastern tropical Indian Ocean (10°S–0°N and 90°E–110°E) [31]. Based on the years 1982–2005, the anomaly is determined relative to a monthly climatological seasonal cycle, and to calculate weekly anomalies, the monthly climatology is linearly interpolated. Weekly DMI was merged into monthly DMI in this study. The positive and negative IOD events can be distinguished based on a DMI threshold of ±0.5 °C. In this study, we not only focused on years that have the highest value of ONI and DMI, but also considered the years that have strong wind stress and EMT to yield more reliable results in terms of coastal upwelling as shown in Table 1.

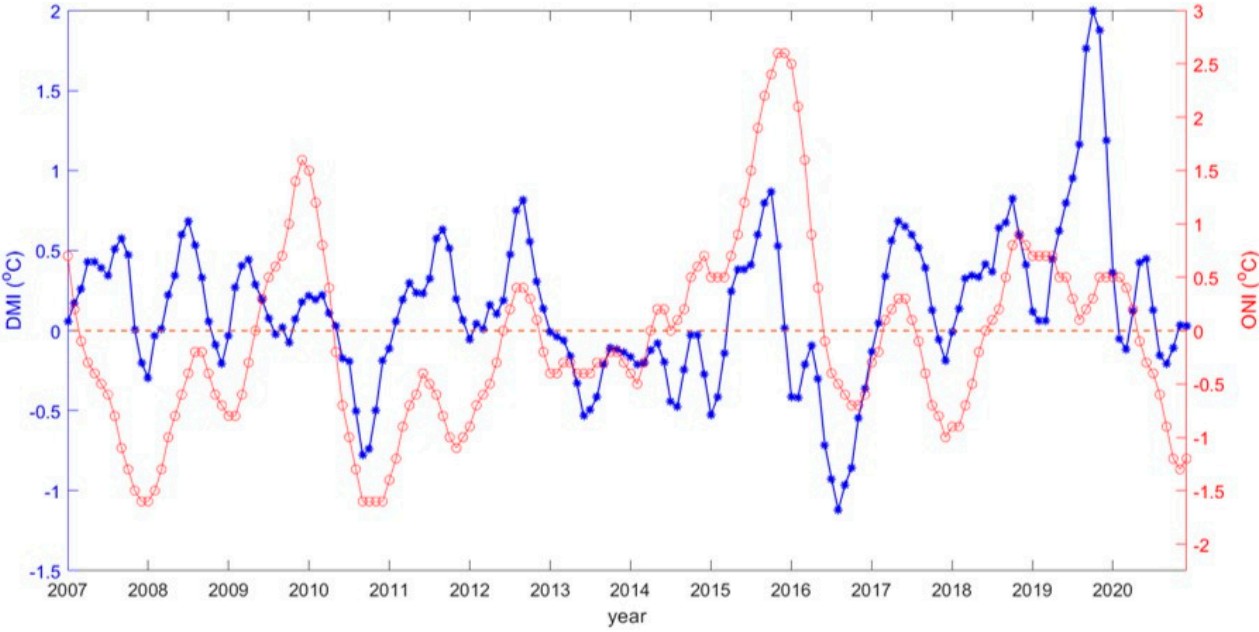

**Figure 2.** Plot of Oceanic Nino Index (ONI) (red line) and Dipole Mode Index (DMI) (blue line) during 2007–2020.

**Table 1.** El Nino-Southern Oscillation (ENSO) and Indian Ocean Dipole (IOD) years.

|  | Positive IOD | Negative IOD | El Nino | La Nina | El Nino + Positive IOD | La Nina + Negative IOD |
|---|---|---|---|---|---|---|
| 1 | **2008** |  |  | **2007** |  |  |
| 2 | 2012 | **2016** | **2009** | 2010 | **2015** | **2010** |
| 3 | 2017 |  | 2015 | 2011 |  |  |
| 4 | 2019 |  |  |  |  |  |

## 3. Results

### 3.1. Spatio-Temporal Variability of chl-a

The monthly climatology mean values of chl-a along the southern coast of LSI are presented in Figure 3. The chl-a concentration during the northwest season (December–February) ranges from 0.12 to 0.16 mg m$^{-3}$ in coastal water regions. During Transitional Season I (March–May), the chl-a concentration slightly increases, ranging from 0.18 to 1 mg m$^{-3}$. Additionally, a prominent increment of chl-a of more than 1.5 mg m$^{-3}$ was found during the southeast monsoon season along the southern coast of Bali, Sumba, and Timor Island. The chl-a concentration is evenly distributed across the southern coast of LSI. During Transitional Season II (September–November), the chl-a concentration gradually decreases and reaches the lowest value in November. The highest chl-a concentration, of more than 1.5 mg m$^{-3}$, occurs in August, whereas the lowest chl-a concentration, about 0.17 mg m$^{-3}$ occurs in February. In

short, the high chl-a concentrations observed during the southeast monsoon season indicate that the upwelling is more pronounced than in the other seasons.

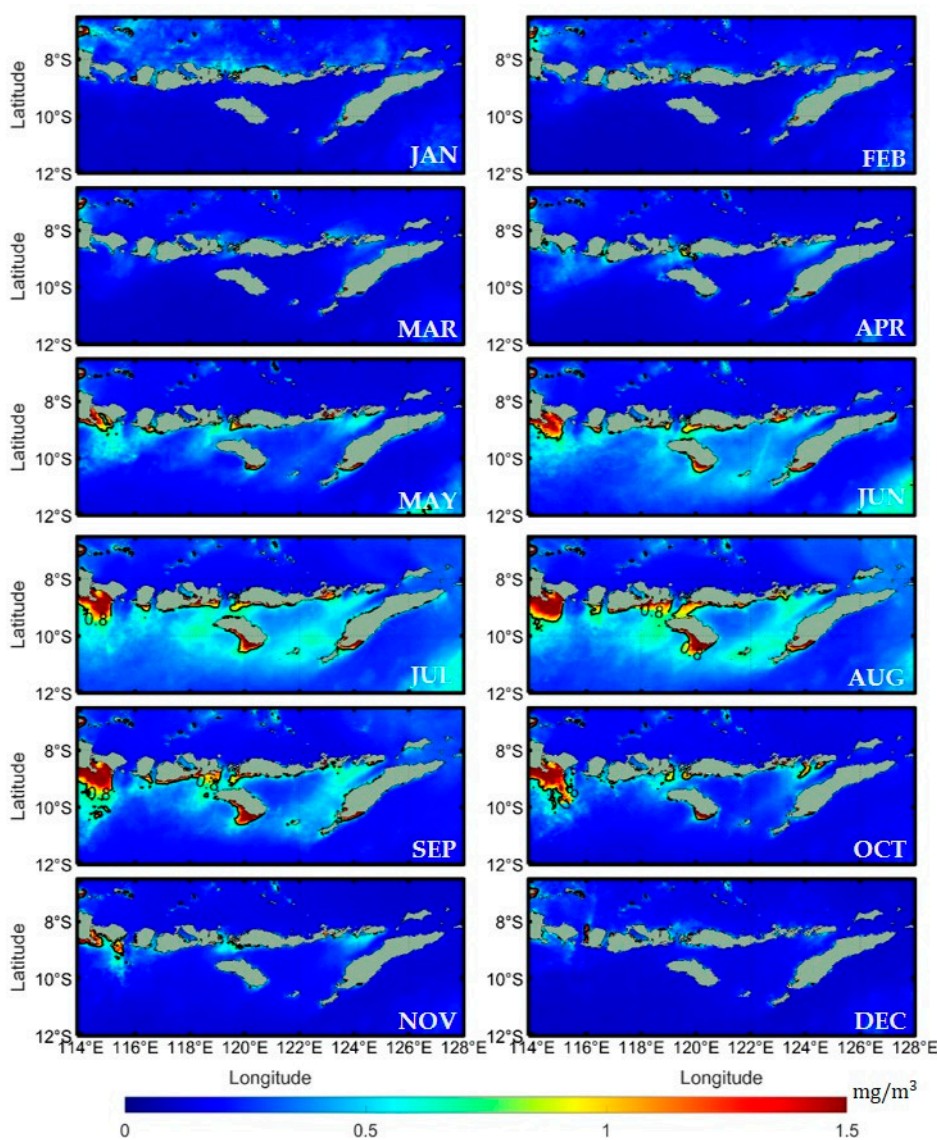

**Figure 3.** The climatological monthly mean values of chlorophyll-a during 2007–2020.

### 3.2. Spatio-Temporal Variability of SST

The monthly climatology mean values of SST along the southern coast of LSI are presented in Figure 4. During the northwest season (December–February), the SST ranged from 28 to 31 °C in coastal water regions. The highest SST (more than 31 °C) occurs in the eastern part of Timor, which spans from 123.6°E to 128°E and from –8.4°S to –12°S. During Transitional Season I (March–May), the distribution of high SST values from 28 to 31 °C decreases. The most significant decrement, by 1–1.5 °C, occurs in May. The lowest SST occurs during the southeast monsoon season (June–August). The low SST values at 25–27 °C are evenly distributed along the coastal water region. In addition, the lowest SST during this season was found at the southern coast of Bali, which is 113.75°E to 114.1°E and –9°S to –9.5°S. During Transitional Season II (September–November), the SST tends to increase, although a low SST (lower than 27.5 °C) still occurs at the southern coast of East Java and Bali. We conclude that low SSTs are evenly distributed at the southern coast of LSI during the southeast monsoon season, which indicates that upwelling is much more robust in this period.

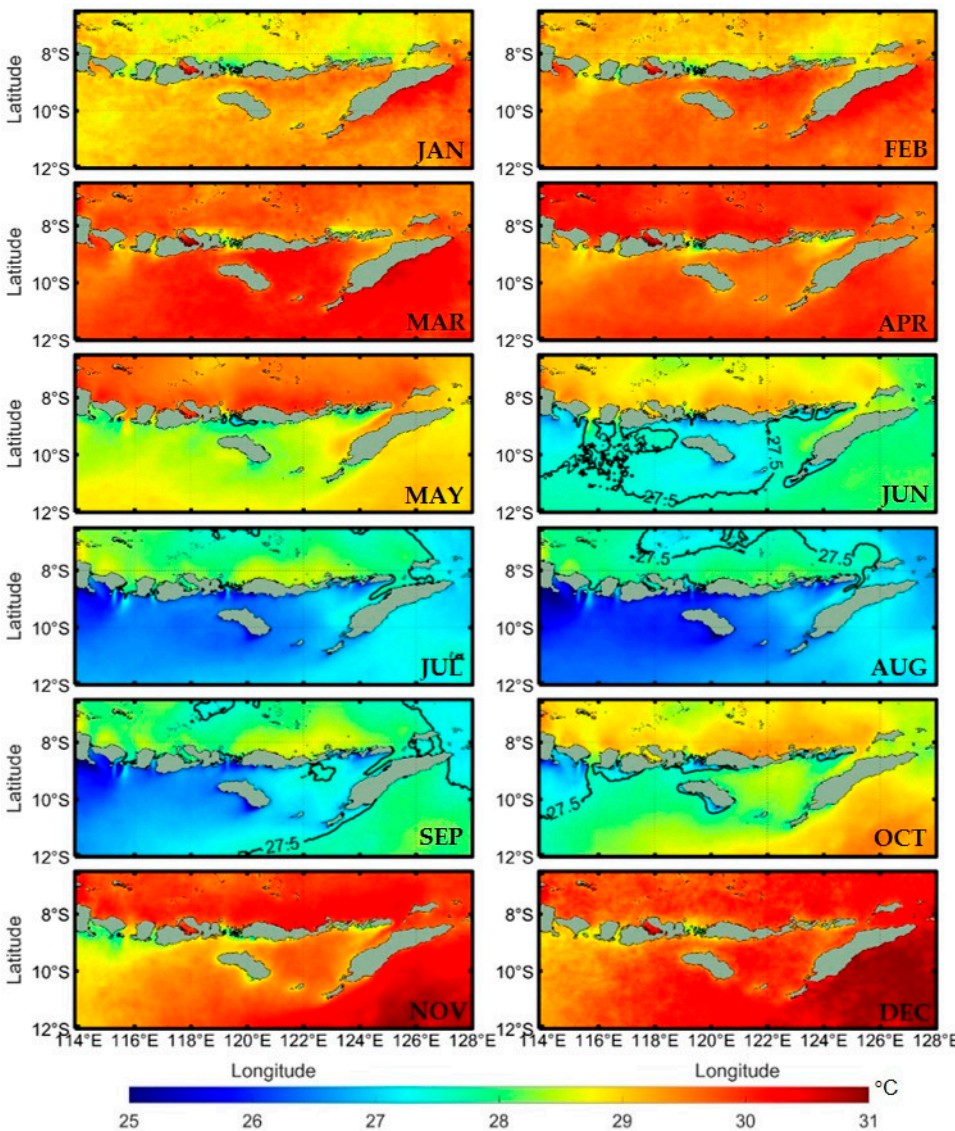

**Figure 4.** The climatological monthly mean values of sea surface temperature (SST) during 2007–2020.

### 3.3. Spatio-Temporal Variability of Wind Stress

The monthly climatology mean values of wind stress along the southern coast of LSI are presented in Figure 5, which denotes a prominent seasonal cycle. During the northwest season (December–February), the wind stress ranges from 0.01 to 0.05 Nm$^{-2}$ in coastal water regions. In January, the most robust wind stress (0.08 Nm$^{-2}$) occurs in the western Savu Sea. During Transitional Season I (March–May), the strong wind stress distribution increases from April to May, whereas the weakest wind stress, lower than 0.01 Nm$^{-2}$, occurs in March. The wind stress significantly increases during the southeast monsoon season (June–August). Strong wind stress (0.05–0.1 Nm$^{-2}$) is evenly distributed along the coastal water region and reaches its maximum value (higher than 0.1 Nm$^{-2}$) in July. During Transitional Season II (September–November), the wind stress tends to decrease, although an intense wind stress (0.05 Nm$^{-2}$) still occurs at the southern coast of East Java and Bali in September. We noticed that the strong wind stress during the southeast monsoon season is in line with low SSTs and high chl-a concentrations, which also can be seen in the same season; this indicates that SST and chl-a concentration in this area are remarkably sensitive to wind forcing. The wind stress intensity and direction control the spatial distribution of SST and chl-a concentration. The strong wind stress likely induces upwelling and lowers the mixed layer depth. As a result, the southern coast of LSI experiences a high chl-a

concentration and a low SST. This suggests that enhanced marine primary productivity along the southern coast of LSI in June–August might be critical for the region's marine life and habitats.

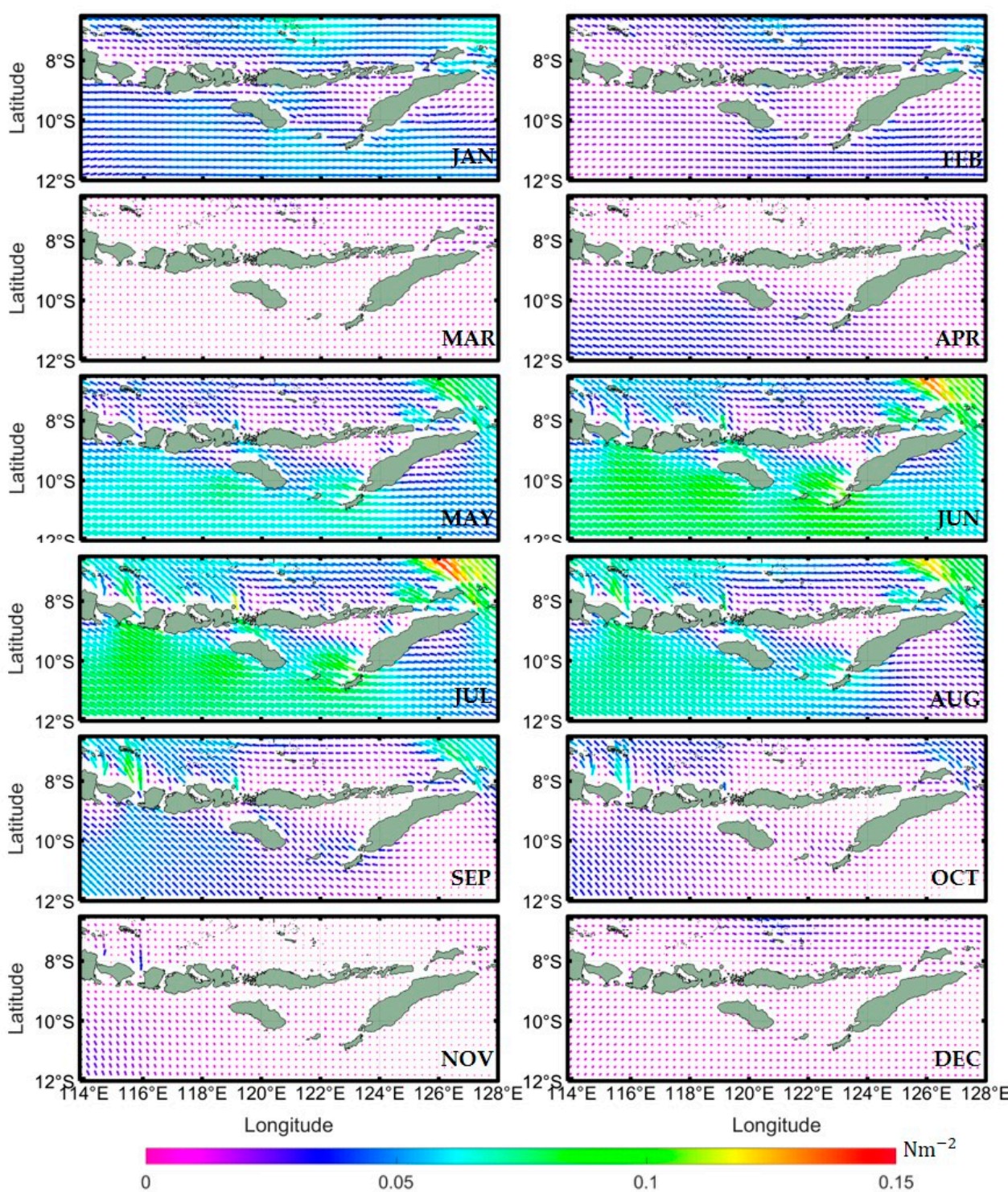

**Figure 5.** The climatological monthly mean values of wind stress during 2007–2020.

### 3.4. The Four Locations with a Unique SST and Surface Wind Characteristics

A map of climatology monthly mean values of SST and surface wind in Indonesia seas is shown in [56]. The results indicate that SST and surface wind distribution are well correlated. Furthermore, Iskandar et al. [57] found that monthly mean values of chl-a and

surface wind off the southern coast of Sumatra and Java corresponded well. In our study, we present the most recent sea surface wind and SST data in the most intense part of the upwelling period, i.e., August, to reveal the more specific association between SST and sea surface wind along the southern coast of LSI (Figure 6), which has been overlooked in earlier research.

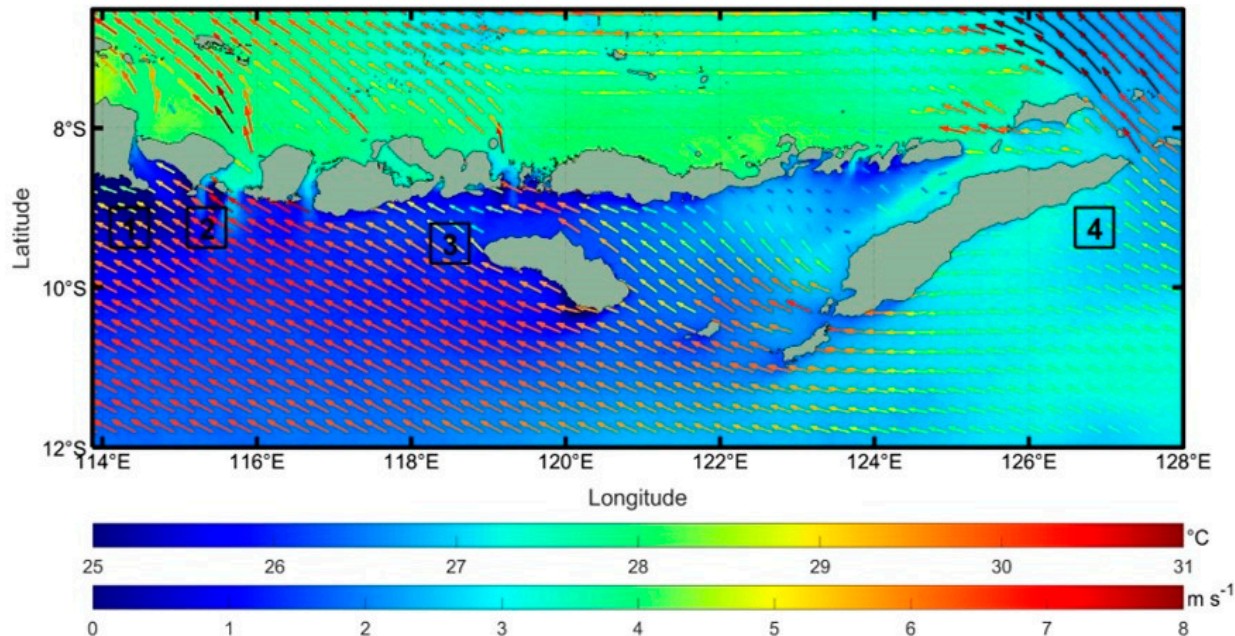

**Figure 6.** The climatological monthly mean values of sea surface wind and sea surface temperature (SST) in August (2007–2020). Shaded portions represent SST. Wind speed is indicated by the length of the arrow and its color.

Cold SSTs (less than 26 °C) are not evenly distributed along the southern coast of LSI. We studied four different areas with different relationships between wind speed and SST. In Box 1, the SST is cooler than in Box 2. In contrast, the wind speed in Box 1 is lower than in Box 2. As a result, there is inconsistency in the relationship between SST and surface wind speed along the southern coast of LSI, where several scientists in various regions of the Indonesian seas have shown that higher wind speeds result in lower SSTs. [21,40–42,58,59]. Additionally, the relationships between SST and wind speed are consistent in Boxes 3 and 4. High wind speeds in Box 3 coincide with low SSTs, whereas low wind speeds in Box 4 are in line with high SSTs.

In Figure 7, we show the mean values of SST and surface wind from the four boxes in Figure 6 to indicate their association in these areas. High SSTs occur two times per year in March and December. The highest SSTs occur in March (29.6 and 29.3 °C) in Boxes 1 and 2, respectively. In contrast, the lowest SSTs occur in August for Box 1 and Box 2 (25.2 and 25.8 °C), respectively. From January to May and from October to November, the SST in Box 2 is lower than in Box 1. On the other hand, from June to September and in December, the SST in Box 2 is higher than in Box 1. In addition, the SST in Box 3 is lower than in Box 4 for all months. In March and December, high SSTs occur twice per year in both boxes. The highest SST occurs in March (29.9 °C) in Box 3 and in December (30.9 °C) in Box 4. In contrast, in August, the lowest SSTs are 26.03 and 27.04 °C in Box 3 and Box 4, respectively. Seasonal variation can be observed clearly in all boxes.

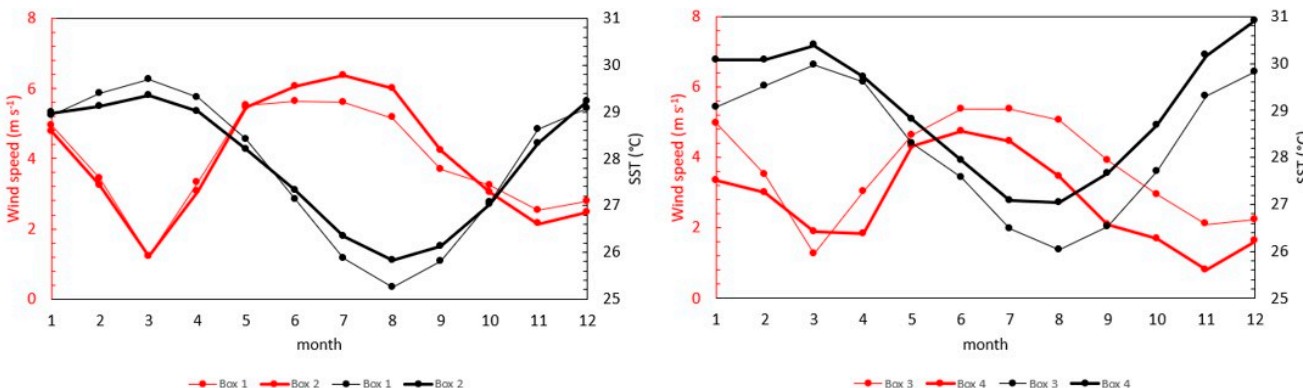

**Figure 7.** The monthly climatology of wind speed and sea surface temperature (SST) within the boxes in Figure 3.

A similar pattern also occurs in the wind speed for all boxes, showing the seasonal variation. In March, the lowest winds speed in for Boxes 1, 2, and 3 are 1.21, 1.22, and 1.22 ms$^{-1}$, respectively. In December, the lowest wind speed in Box 4 is 1.89 ms$^{-1}$. In July, the maximum wind speeds in Boxes 1, 2, and 3 are 5.6, 6.3, and 5.3 ms$^{-1}$, respectively. Additionally, the maximum wind speed (4.7 ms$^{-1}$) in Box 4 occurs in June. During the southeast monsoon season (June–August), we noticed that the SST in Box 1 is about 0.5 °C lower than in Box 2, whereas the wind speed in Box 1 is about 1 m s$^{-1}$ lower than in Box 2. During the southeast monsoon season, the relationship between wind speed and SST becomes inconsistent. To analyze this discrepancy, we examined Ekman dynamics, as explained in the next section.

We also noticed that some areas along the southern coast of East Java and Bali have low SST during the northwest monsoon season. This may be caused by the surface heat flux [43]. The high wind speed induced a latent heat release, which caused low SSTs during the northwest monsoon season. The variation in latent heat flux is generated by the wind speed, which regulates the SST variation along the Java Sea [60]. We also noticed that onshore EMT occurs during the northwest monsoon season, as shown in Figure 8. In addition, the positive EPV indicates that downwelling is dominant along the southern coast of East Java and Bali during the northwest monsoon season, as shown in Figure 9. Therefore, the EPV and EMT are not responsible for low SSTs in these regions.

### 3.5. Spatio−Temporal Variability of EMT

The results of the monthly climatology mean values of EMT are presented in Figure 8. Onshore EMTs occur from December to March, reaching maximum and minimum values in January and March (about 2 and 0.1 m$^2$s$^{-1}$) along the southern coast of LSI, respectively. We also noticed that the onshore EMT occurs under northwesterly wind conditions shown in [57,61]. Additionally, the offshore EMT is dominant from April to November. The highest offshore EMT, of more than 3 m$^2$s$^{-1}$, occurs in July along the southern coast of Bali, Lombok, and Sumbawa. On the other hand, the weakest offshore EMT, lower than 1 m$^2$s$^{-1}$, occurs in November before it changes into an onshore EMT in December.

Additionally, the transition from onshore EMT to offshore EMT occurs from March to April. The intense offshore EMT observed during the southeast monsoon season is caused by the high wind stress in the same period. The most prominent offshore EMT occurs along the southern coast of Bali, Lombok, Sumbawa, Sumba, and Timor during the southeast monsoon season. Through the mechanism of coastal upwelling, a strong offshore EMT helps the upwelling to lift cold and rich-nutrient water from the subsurface to the ocean surface. As a consequence, a large distribution of high chl-a concentrations and a massive low SST distribution occur during the southeast monsoon season along the southern coast of LSI.

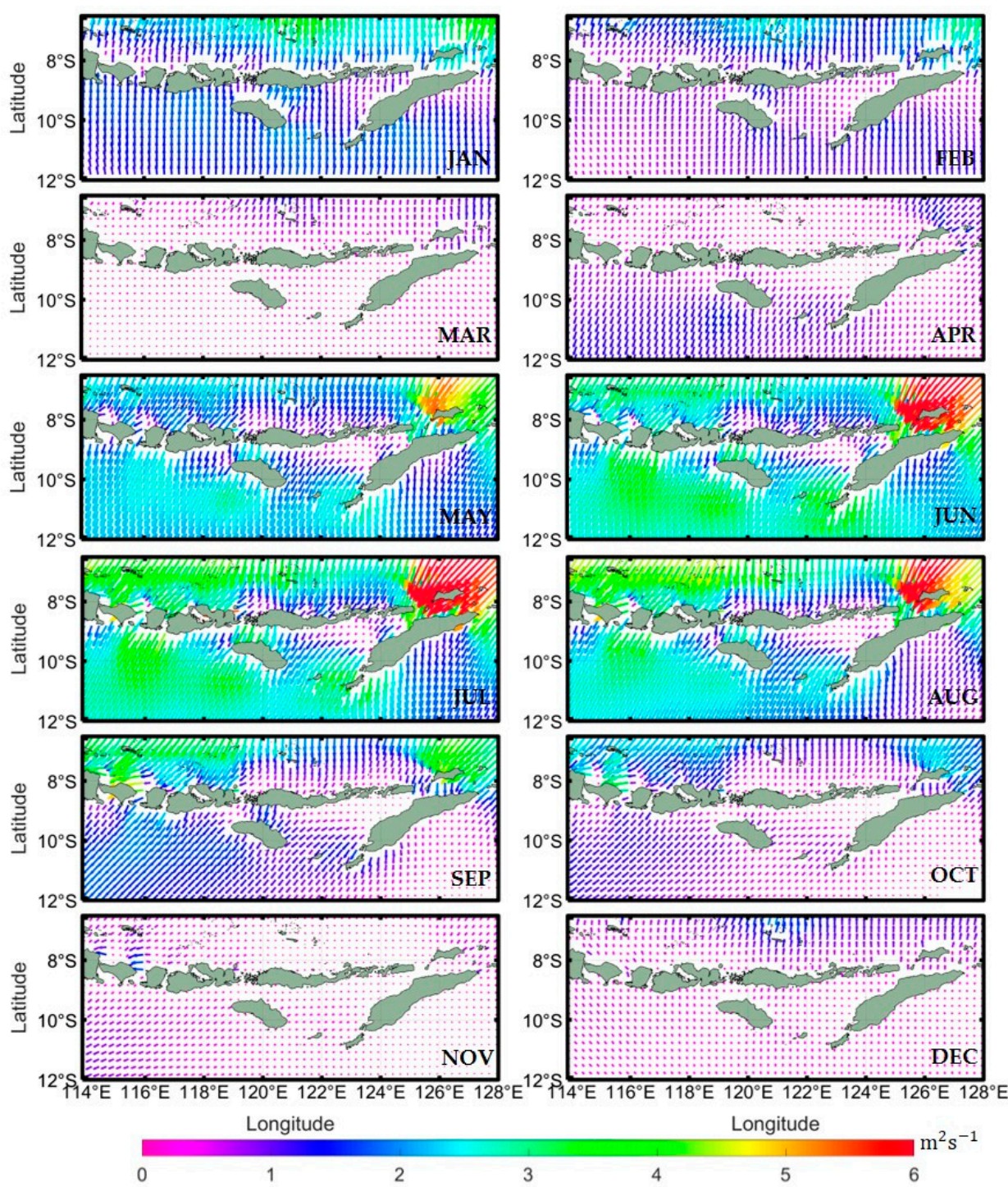

**Figure 8.** The climatological monthly mean values of Ekman Mass Transport (EMT) during 2007–2020.

### 3.6. Spatio-Temporal Variability of EPV

The results of the monthly climatology mean values of EPV are presented in Figure 9. Positive and negative EPVs indicate downwelling and upwelling, respectively. During the northwest monsoon season (December–February), the EPVs are dominated by positive EPV values from $0.5 \times 10^{-5}$ to $1.5 \times 10^{-5}$ ms$^{-1}$ along the southern coast of LSI. Additionally, during Transitional Season I (March–May), the distribution of positive EPVs decreases, and EPV values become negative in April and May along the southern coast of Java, and the western coasts of Sumba and Sumbawa. The negative EPVs are intensified in these regions during the southeast monsoon season and reach maximum values, of more

than $-1.5 \times 10^{-5}$ ms$^{-1}$, in July. During Transitional Season II (September–November), the negative EPVs start to decrease in these three regions and, in December, change into positive EPVs.

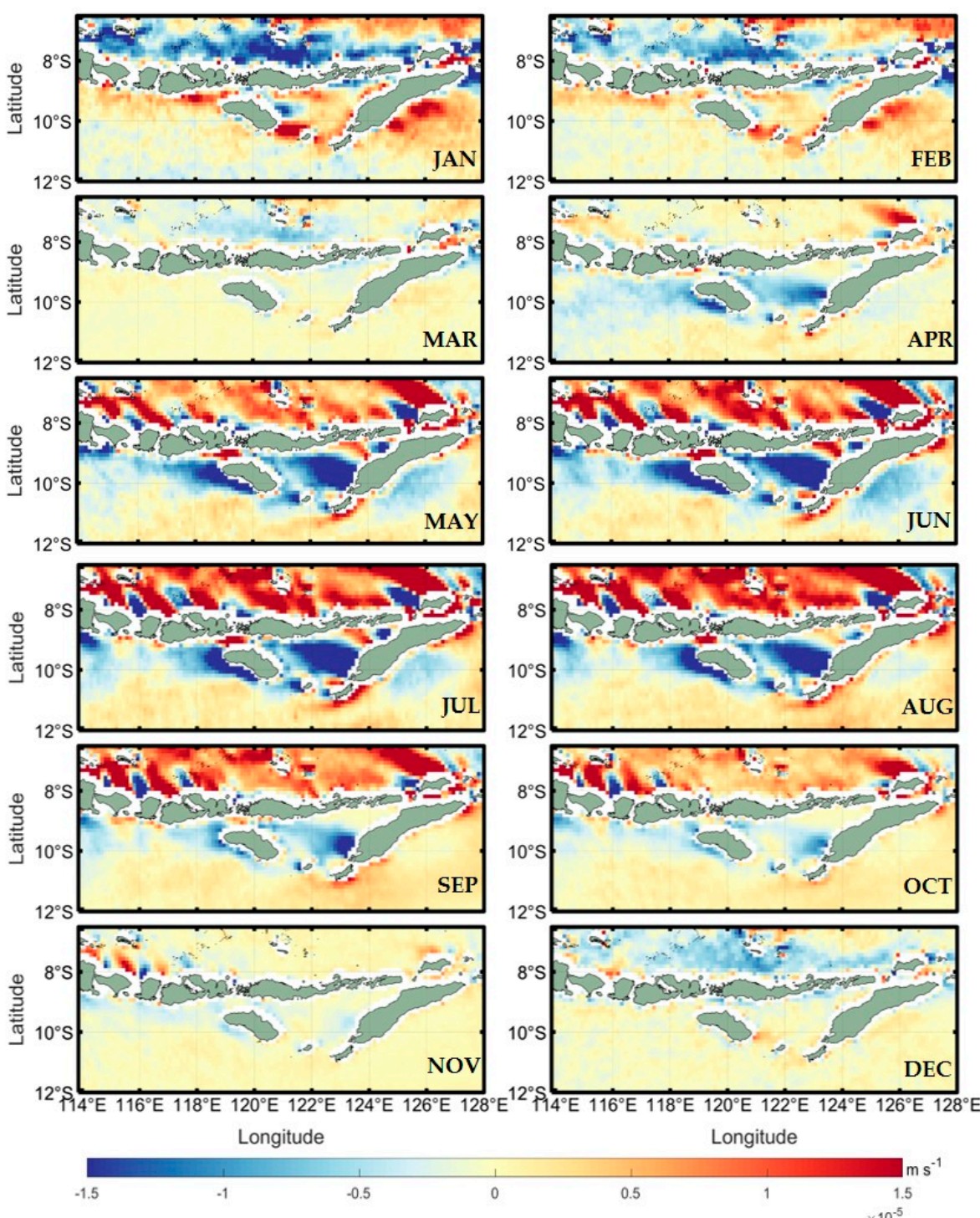

**Figure 9.** The climatological monthly mean values of Ekman Pumping Velocity (EPV) during 2007–2020.

We used an average of the EMT meridional component and the EPV in all boxes to examine the relationship between EMT, EPV, and the spatial distribution of SST, as shown in Figure 10. As discussed in the previous section, from January to May and from October to November, the SST in Box 2 is lower than in Box 1. On the other hand, from

June to September and in December, the SST in Box 2 is higher than in Box 1. From January to May, the meridional EMT values in Box 1 and 2 are almost equal. After May, the negative meridional component of EMT in Box 2, which indicates the offshore EMT, becomes stronger than in Box 1. The maximum difference in the negative meridional value between Box 1 and Box 2 (of 0.621 m$^2$s$^{-1}$) occurs in July.

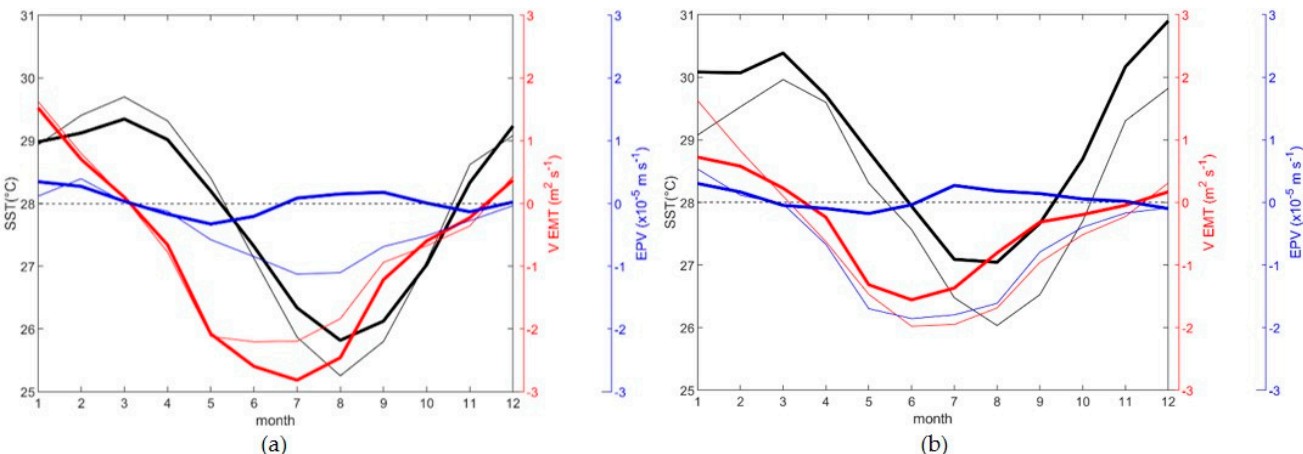

**Figure 10.** The climatological mean values of sea surface temperature (SST) (black), Ekman Mass Transport (EMT) meridional component (red), and Ekman Pumping Velocity (EPV) (blue). Thin and thick lines denote the mean value in (**a**) Box 1 and Box 2; (**b**) Box 3 and Box 4, respectively.

The maximum difference of the EPV ($-1.253 \times 10^{-5}$ ms$^{-1}$) also occurs in August between Box 1 and Box 2. EPV becomes positive in Box 2 during the southeast monsoon season, whereas EPV is negative in Box 1. As a result, Box 1 is characterized by a negative EPV and a weak offshore EMT. Conversely, Box 2 is characterized by a strong offshore EMT and a positive EPV. A negative EPV enhances EMT-produced upwelling, despite the fact that coastal upwelling induced by an offshore EMT at Box 1 is weak. Positive EPV-induced downwelling at Box 2, by comparison, counteracts the robust offshore EMT-induced upwelling.

The SST in Box 3 is lower than in Box 4 in all months. The meridional EMT has a positive value from January to March, indicating the onshore EMT in Box 3 and Box 4. After March, the negative meridional component of EMT in Box 3, which indicates an offshore EMT, becomes stronger than in Box 4. The maximum difference of the negative meridional value between Box 3 and Box 4 ($-0.88$ m$^2$ s$^{-1}$) occurs in August. In addition, the maximum difference of the EPV ($-1.253 \times 10^{-5}$ ms$^{-1}$) also occurs in August between Box 3 and Box 4. We conclude that Box 3 is delineated by a negative EPV and a strong offshore EMT during the southeast monsoon season.

In contrast, Box 4 is characterized by a weak offshore EMT and a positive EPV during the southeast monsoon. This result suggests that coastal upwelling is induced by the offshore EMT and the negative EPV in Box 3. Although the intensity of a negative EPV indicates that upwelling in Box 3 is stronger than in Box 1 and Box 2, the offshore EMT is not as strong as in Box 1 and Box 2. Therefore, a weak offshore EMT counteracts the upwelling generated by a negative EPV. As a result, the SST in Box 3 is higher than in Box 1 and Box 2. In contrast, the combination of a weak offshore EMT and a positive EPV results in a high SST in Box 4.

Figure 11 depicts the upwelling or downwelling processes in all boxes as a variation in sea water density in vertical profiles derived from reanalysis data. The lifting of denser water masses indicates upwelling, which occurs in all boxes during the southeast monsoon season. From August to September, it can also be seen that the upwelling in Box 2 is weaker than in Box 1, as evidenced by a lower surface density (more than 22.2 kg m$^{-3}$). In addition, the lifting of denser water (24–25 kg m$^{-3}$) in Box 1 is more pronounced than in

the other boxes. This result also coincides with the lowest SST observed in August in Box 1. Furthermore, the lifting of denser water masses in Box 2 seems stronger than in Box 1 during the southeast monsoon season, as shown by the higher surface density (an isopycnal of 22.2 kg m$^{-3}$). However, we noticed that the lifting of denser water (24–25 kg m$^{-3}$) in Box 2 is not stronger than in Box 1. In contrast, the upwelling in Boxes 3 and 4 is not as strong as in Boxes 1 and 2 because the isopycnal of 22.2 kg m$^{-3}$ does not reach the ocean surface. We conclude that the lowest SST in Box 1 is generated by the combination of a negative EPV and a weak offshore EMT. In contrast, the highest SST in Box 4 is due to the combination of a positive EPV and a weak offshore EMT during the southeast monsoon season. The overall results suggest that each area has a different mechanism that may affect the distribution of SST.

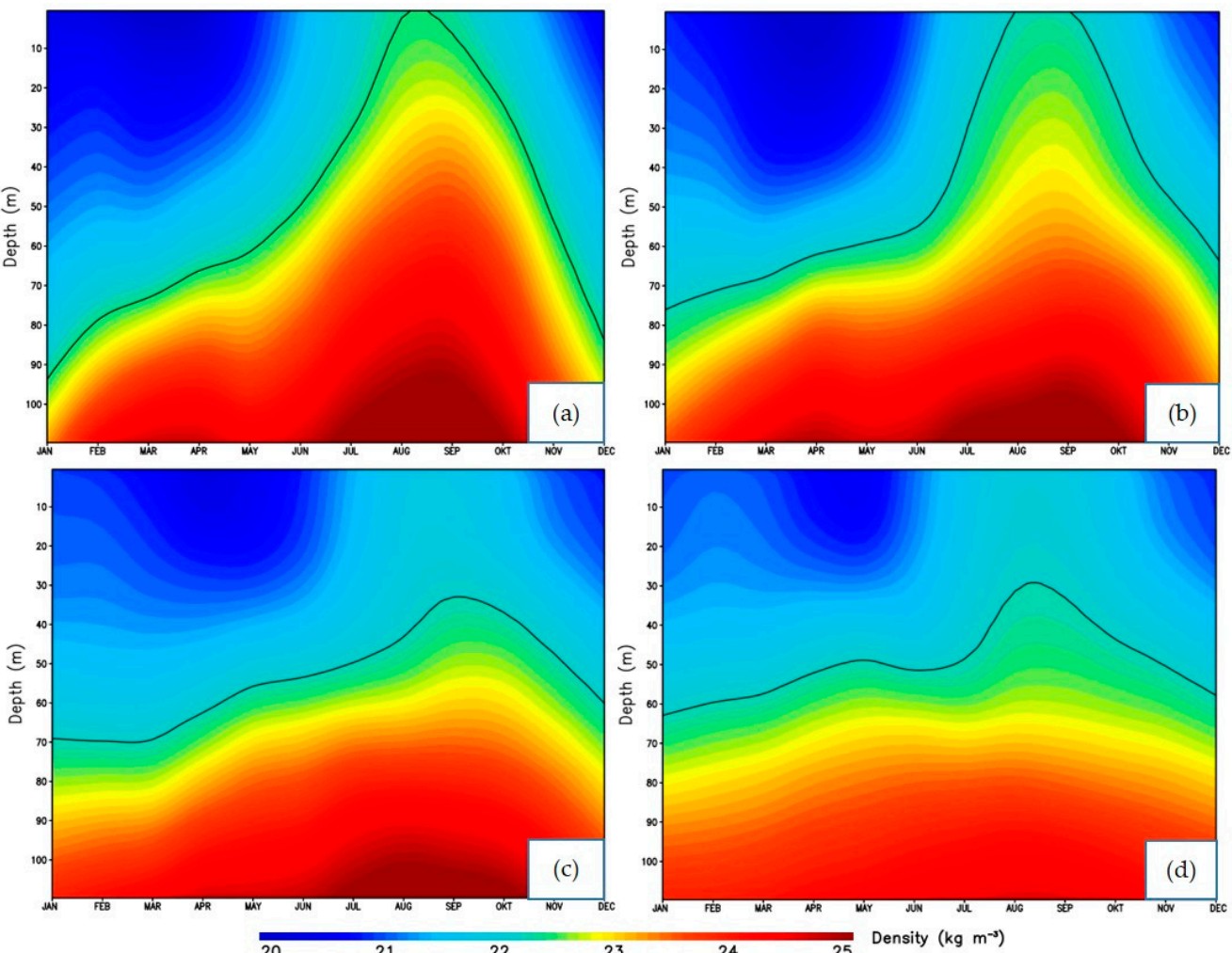

**Figure 11.** The climatological monthly mean values of the sea water density in the vertical profile in (**a**) Box 1, (**b**) Box 2, (**c**) Box 3, and (**d**) Box 4, respectively. The black contours are isopycnals of 22.2 kg m$^{-3}$.

We also calculated the mixed layer depth based on available Argo data along the southern coast of LSI (Figure 12). During the southeast monsoon season, the mixed layer depths are shallower than those during the northwest monsoon season. Furthermore, the mixed layer depth during the southeast monsoon season ranges from 13 to 35 m, and the mixed layer depth during the northwest monsoon season ranges from 30 to 54 m. This indicates that the upwelling during the northwest monsoon season is weaker than during the southeast monsoon season. The shallow mixed layer depth, compared with a deeper mixed layer depth, allows the phytoplankton to grow faster as the phytoplankton can remain closer to the surface and, as a result, receive more light. Additionally, the

thermocline deepens, preventing nutrient-rich water from being entrained to the surface and resulting in a decrease in cold and nutrient-rich water along the southern coast of LSI.

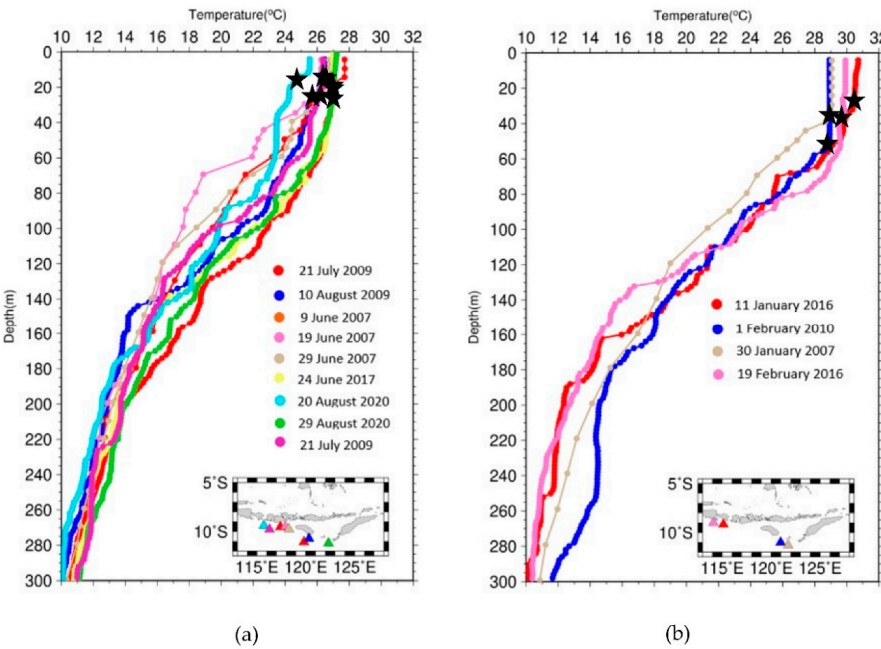

(a)                                                                          (b)

**Figure 12.** Argo float data during (**a**) the southeast monsoon season and (**b**) the northwest monsoon season. The black star symbols indicate the calculated values of mixed layer depth.

As indicated in Table 2, we performed a statistical assessment to determine the relation between the monthly climatology of SST, EPV, and EMT in all boxes from 2007 to 2020. SST and EPV have a higher correlation than SST and meridional EMT in Box 1, by 0.61 and −0.58, respectively. This result suggests that the impact of EMT on SST variability is weaker than that of EPV. In contrast, SST and EMT have a higher correlation than SST and EPV in Box 3, which means that the impact of EPV on SST variation is weaker than that of EMT in Box 3. A negative correlation between EMT and SST in all boxes means that as the EMT value increases, the SST value decreases, and vice versa. This indicates that the stronger the EMT, the cooler the SST. In contrast, a positive correlation between EPV and SST indicates that, as the EPV increases, SST increases, and vice versa. This indicates that the stronger the EPV, the warmer the SST. The overall results imply that EMT and EPV play an important role in defining the SST along the southern coast of LSI. We also determined multiple correlations between EMT and EPV on SST variation to determine the impact of both EMT and EPV on the ocean surface conditions. The multiple correlation values are higher than the single correlation values in all boxes. The highest multiple correlation value (0.71) is in Box 1, followed by 0.68 in Box 3, 0.57 in Box 2, and 0.51 in Box 4. These findings suggest that the combined effects of EMT and EPV play a crucial role in defining SST variation along the southern coast of LSI.

**Table 2.** Correlation between monthly averages of sea surface temperature (SST), Ekman Mass Transport (EMT), and Ekman Pumping Velocity (EPV) in all boxes from 2007 to 2020.

|       | Variable | EMT   | EPV   | EMT and EPV |
|-------|----------|-------|-------|-------------|
| SST   | Box 1    | −0.58 | 0.61  | 0.71        |
|       | Box 2    | −0.57 | −0.12 | 0.57        |
|       | Box 3    | −0.61 | 0.58  | 0.68        |
|       | Box 4    | −0.5  | −0.14 | 0.51        |

*3.7. Interannual Variability*

3.7.1. The Effects of the 2009 El Nino and the 2007 La Nina

The variability in chl-a, SST, wind stress, and Ekman dynamics during the southeast monsoon season in the 2009 El Nino and the 2007 La Nina is represented in Figures 13 and 14, respectively. During the 2009 El Nino, the positive anomaly of SST and the negative anomaly of chl-a were dominant along the southern coast of LSI. In contrast, the upwelling strength during the 2004 El Nino was stronger than that in the normal years (2000 and 2001), which is indicated by more positive (negative) anomaly values of chl-a (SST), as reported by [11]. This may have happened due to the shoaling thermocline depths, which are favorable upwelling conditions [30]. However, in this case, the positive anomaly of SST and the negative anomaly of chl-a can be understood as being a result of the negative anomalies of wind stress and EMT.

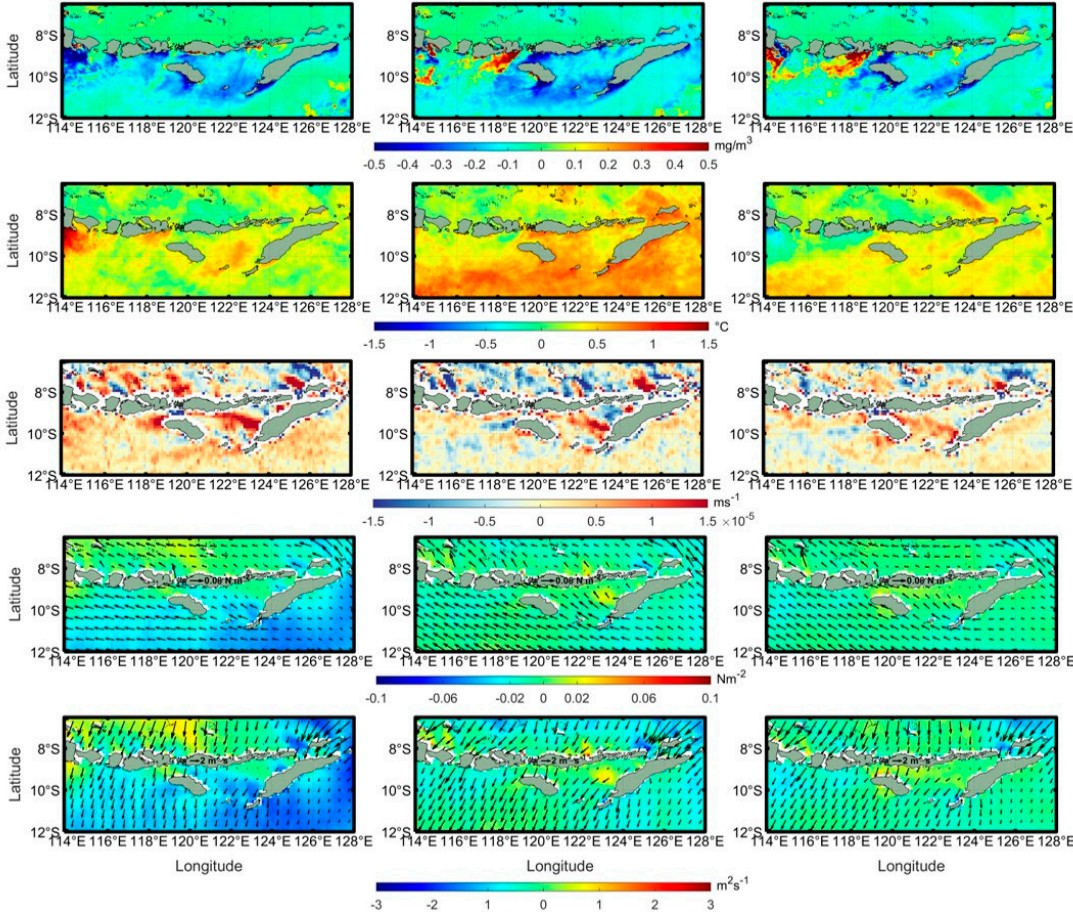

**Figure 13.** The anomalies of chlorophyll-a, sea surface temperature (SST), Ekman Pumping Velocity (EPV), wind stress, and Ekman Mass Transport (EMT) during the 2009 El Nino (June–August from left to right), respectively (from top to bottom).

Additionally, during the 2007 La Nina, we observed that the positive chl-a anomaly and the negative SST anomaly were prominent along the southern coast of LSI. A contrary result has been found in the previous studies [11,40,42]. La Nina induced a negative chl-a concentration and a positive SST anomaly as the Indonesian throughflow (ITF) brings warm water from the Pacific Ocean to the Indian Ocean via the Lombok strait and other straits in the LSI and deepens the thermocline layer [30]. Hence, upwelling intensity is reduced during a La Nina event. The positive chl-a anomaly and the negative SST anomaly during the 2007 La Nina also coincided with positive wind stress and EMT anomalies. These results indicate that wind stress and EMT play a crucial role in defining the ocean surface conditions along the southern coast of LSI.

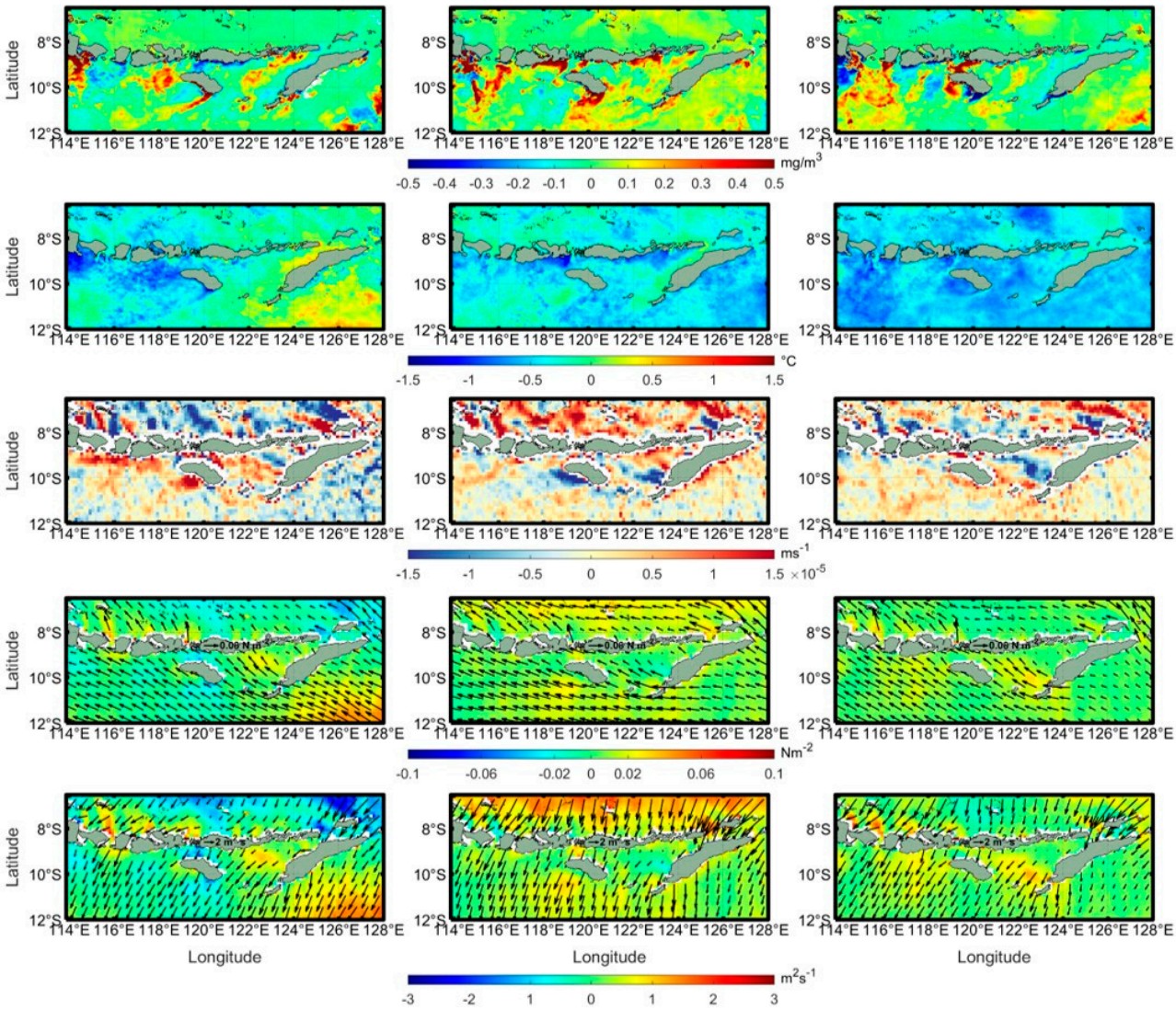

**Figure 14.** The anomalies of chlorophyll-a, sea surface temperature (SST), Ekman Pumping Velocity (EPV), wind stress, and Ekman Mass Transport (EMT) during the 2007 La Nina (June–August from left to right), respectively (from top to bottom).

### 3.7.2. The Effects of the 2008 Positive IOD and the 2016 Negative IOD

The variability in chl-a, SST, wind stress, and Ekman dynamics during the southeast monsoon season in the 2008 positive IOD and the 2016 negative IOD is represented in Figures 15 and 16, respectively. During the 2008 positive IOD, the negative anomaly of SST and the positive anomaly of chl-a were dominant along the southern coast of LSI. The negative anomaly of SST and the positive anomaly of chl-a also coincide with the positive wind stress and EMT anomalies. A similar result was also found in a previous study [41]. A strong wind stress in June–September 2015 was found when the positive IOD and El Nino coincided with each other in the same period. The strong wind stress induced the positive chl-a anomaly caused by the enhanced vertical mixing of the water column associated with a cold SST. In this study, we suggest that wind stress and a stronger than normal offshore EMT induces a positive chl-a anomaly and a negative SST anomaly through the coastal upwelling mechanism. The offshore EMT carries the water mass away from the coastal area, and is then replaced by cold and rich-nutrient water from a subsurface layer. As a result, the study area is surrounded by notable nutrient sources that can cause phytoplankton blooms.

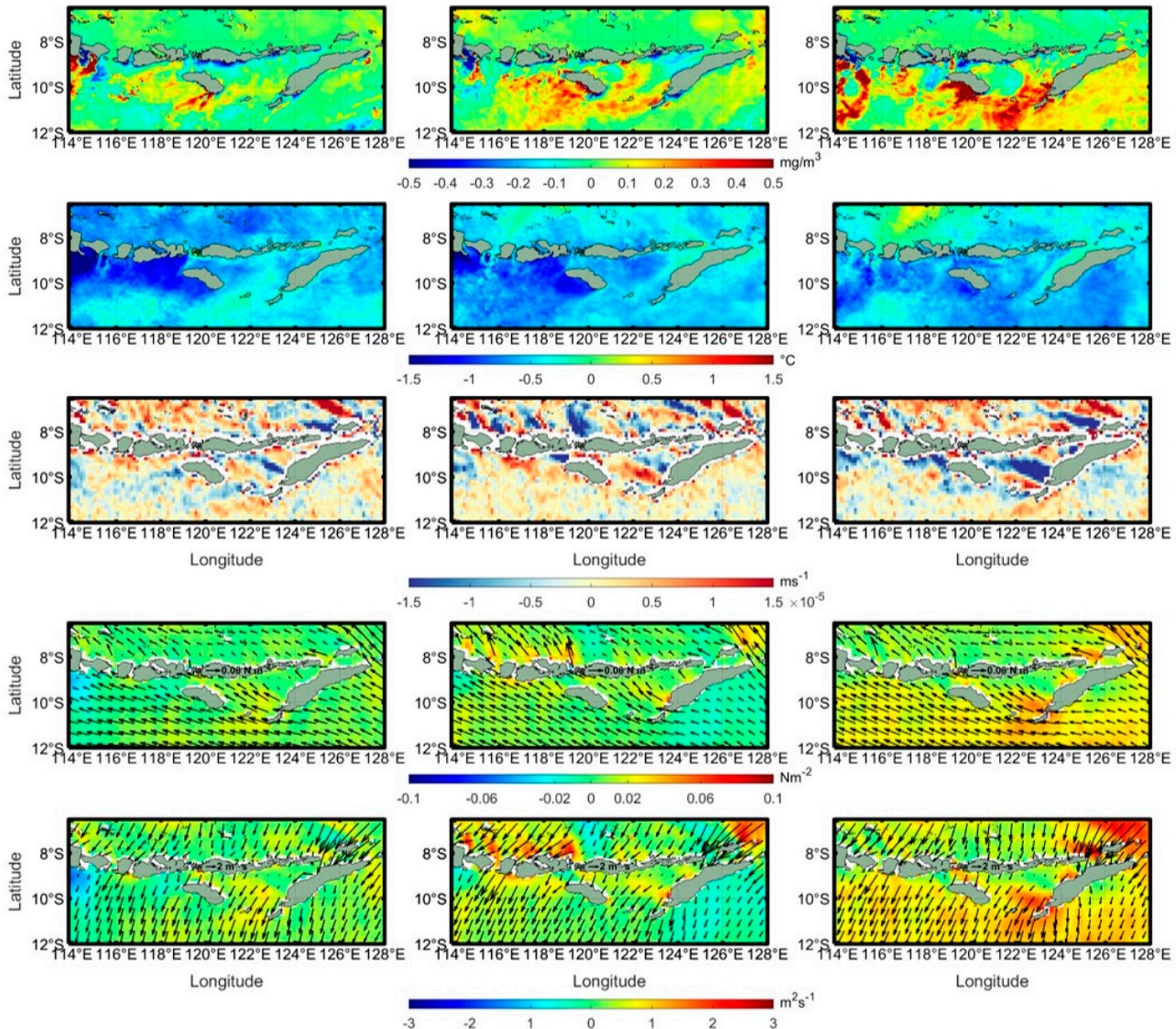

**Figure 15.** The anomalies of chlorophyll-a, sea surface temperature (SST), Ekman Pumping Velocity (EPV) wind stress, and Ekman Mass Transport (EMT) during the 2008 positive Indian Ocean Dipole (IOD) (June–August from left to right), respectively (from top to bottom).

Additionally, during the 2016 negative IOD, the negative chl-a anomaly and the positive SST anomaly were more prominent along the southern coast of LSI. The highest positive SST anomaly (1.5 °C) occurred in June 2016 along the southern coast of Bali. Both the positive SST anomaly and the negative chl-a anomaly were also followed by negative wind stress and EMT anomalies. In contrast, positive wind stress (0.02 $Nm^{-2}$) and EMT (1.5 $m^2s^{-1}$) anomalies occurred along the southern coast of East Java, and the eastern coasts of Sumba and Timor, in July 2016. However, both SST and chl-a concentrations had positive and negative anomaly values in these areas. Furthermore, we noticed a positive EPV value at the same time in these areas, which indicates downward water mass motion (downwelling). These results suggest that the variability in SST and chl-a along these areas are controlled by the wind stress curl (represented by EPV) rather than EMT.

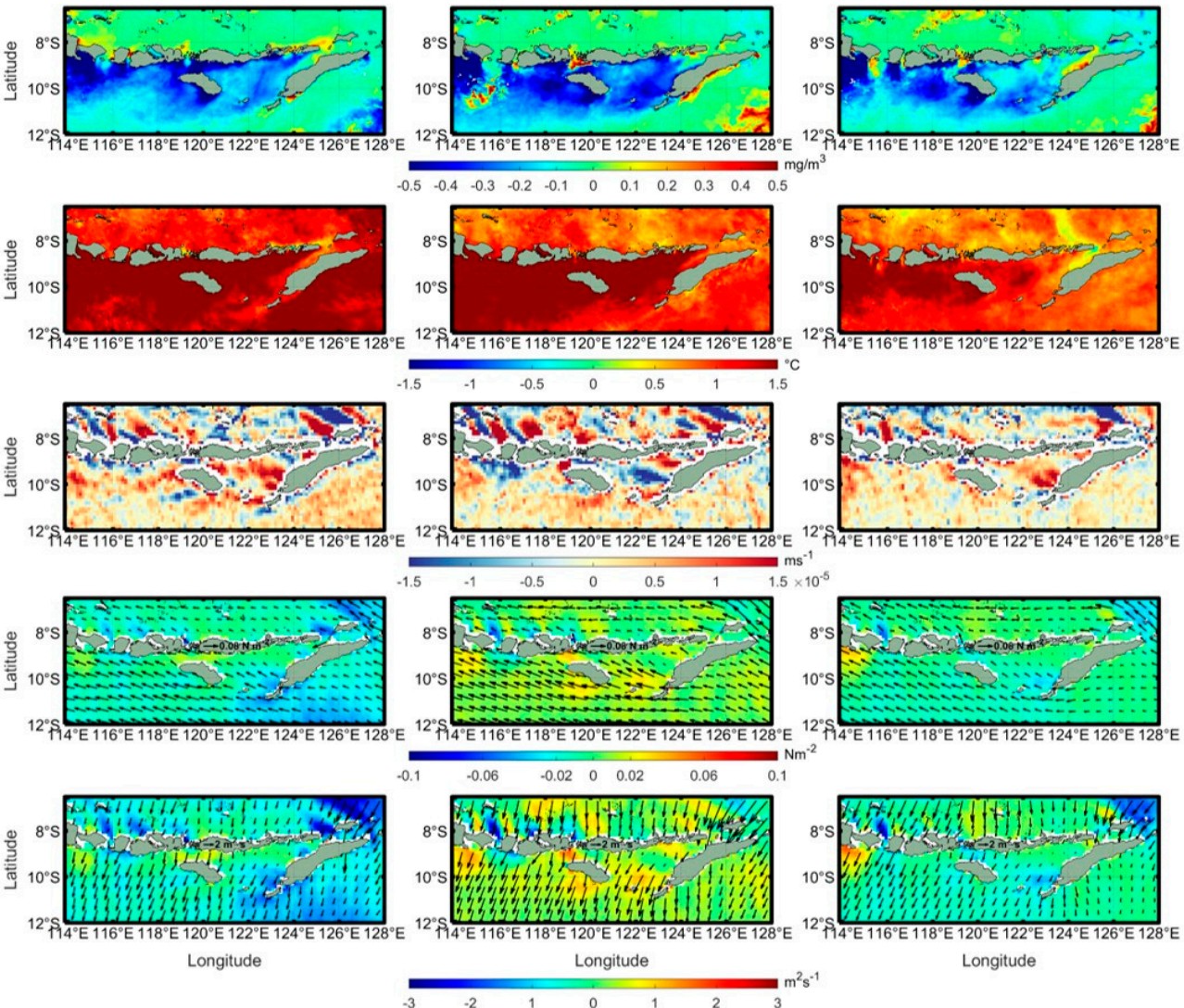

**Figure 16.** The anomalies of chlorophyll-a, sea surface temperature (SST), Ekman Pumping Velocity (EPV), wind stress, and Ekman Mass Transport (EMT) during the 2016 negative Indian Ocean Dipole (IOD) (June–August from left to right), respectively (from top to bottom).

3.7.3. The Effects of the 2015 Positive IOD and El Nino and the 2010 Negative IOD and La Nina

The variability in chl-a, SST, wind stress, and Ekman dynamics during the southeast monsoon season in the 2015 positive IOD and El Nino and in the 2010 negative IOD and La Nina is represented in Figures 17 and 18, respectively. During the 2015 positive IOD and El Nino, the positive anomaly of chl-a and the negative anomaly of SST were dominant along the southern coast of LSI. The positive chl-a anomaly and the negative SST anomaly were not spatially uniform along the southern coast of LSI. This pattern is also seen in the EPV, wind stress, and EMT anomalies. We expected the large positive chl-a anomaly and the negative SST anomaly to be evenly distributed along the southern coast of LSI, as shown by a previous study in other adjacent seas [40]. Our findings reveal that weak wind stress and EMT values during this period are the main reason for this issue.

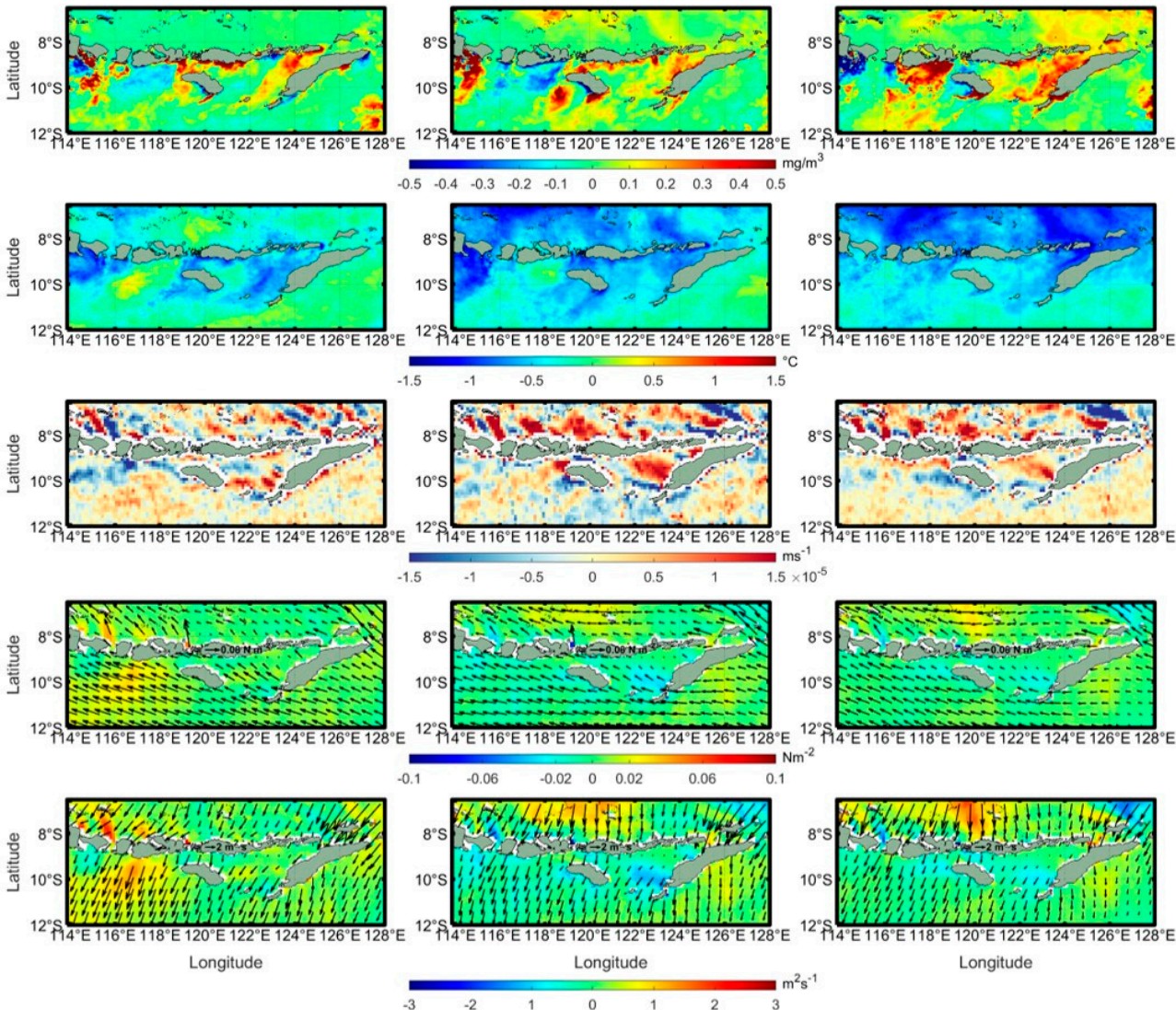

**Figure 17.** The anomalies of chlorophyll-a, sea surface temperature (SST), Ekman Pumping Velocity (EPV), wind stress, and Ekman Mass Transport (EMT) during the 2015 positive Indian Ocean Dipole (IOD) and El Nino (June–August from left to right), respectively (from top to bottom).

We also noticed the same pattern during the 2010 negative IOD and La Nina. The negative anomaly of chl-a and the positive anomaly of SST were dominant along the southern coast of LSI. Additionally, weak wind stress and EMT were not evenly distributed in the same period. In June 2010, positive wind stress and EMT anomalies (0.02 $Nm^{-2}$ and 1.5 $m^2s^{-1}$) were prominent along the Savu Sea and the eastern coast of East Nusa Tenggara and Timor, respectively. The positive wind stress and EMT anomalies occurred in July 2010 along the Savu Sea and the eastern coast of East Nusa Tenggara and Timor. As a result, the positive chl-a anomaly and the negative SST anomaly occurred in July 2010. Additionally, the negative wind stress and EMT anomalies occurred along the coast from Java to Lombok in July 2010, which was also in line with the negative chl-a anomaly and the positive SST anomaly. Furthermore, the spatial distribution of the positive EPV anomaly, which indicates downwelling off the southern coast of Bali to Sumbawa, was more prominent. As a consequence, positive SST and negative chl-a anomalies predominates in these regions.

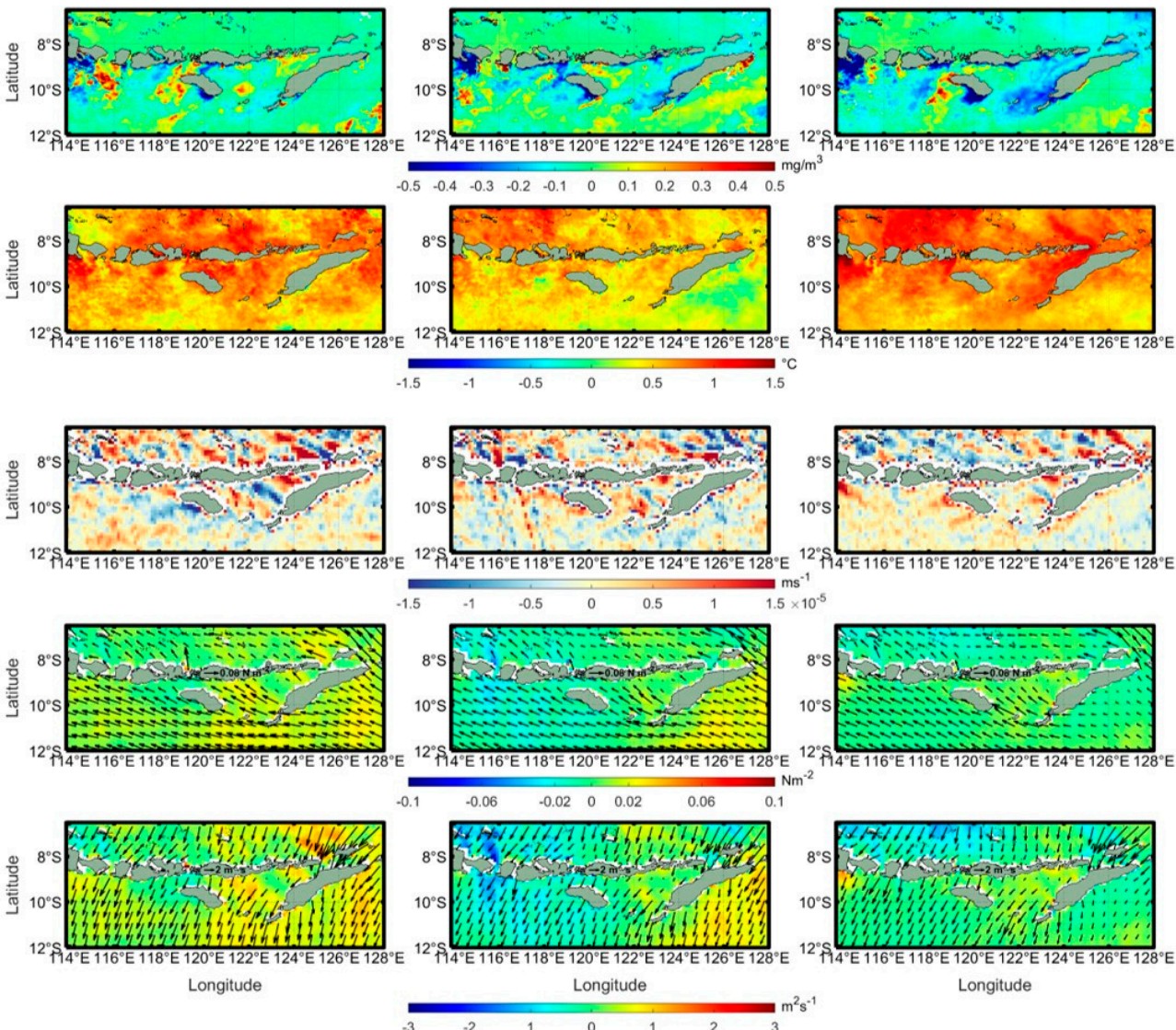

**Figure 18.** The anomalies of chlorophyll-a, sea surface temperature (SST), Ekman Pumping Velocity (EPV), wind stress, and Ekman Mass Transport (EMT) during the 2010 negative Indian Ocean Dipole (IOD) and La Nina (June–August from left to right), respectively (from top to bottom).

### 3.7.4. The Variation of Mixed Layer Depth

The variation in the mixed layer depth in a different phase of ENSO and IOD during the southeast monsoon season is shown in Figure 19. In addition, we also determine the mixed layer depth in each box as shown in Figure 20. The mixed layer depth varies in response to different phases of ENSO and IOD. The mixed layer depth ranged from 70 to 220 m during 2007–2020 along the southern coast of LSI. The shallowest mixed layer depth occurred in June during the 2007 La Nina event, and ranged from 77 to 125 m in all boxes, as shown in Table 3. These results are also in line with the positive anomaly of chl-a concentration and the negative anomaly of SST in all boxes, which correspond to the positive anomalies of wind stress and EMT.

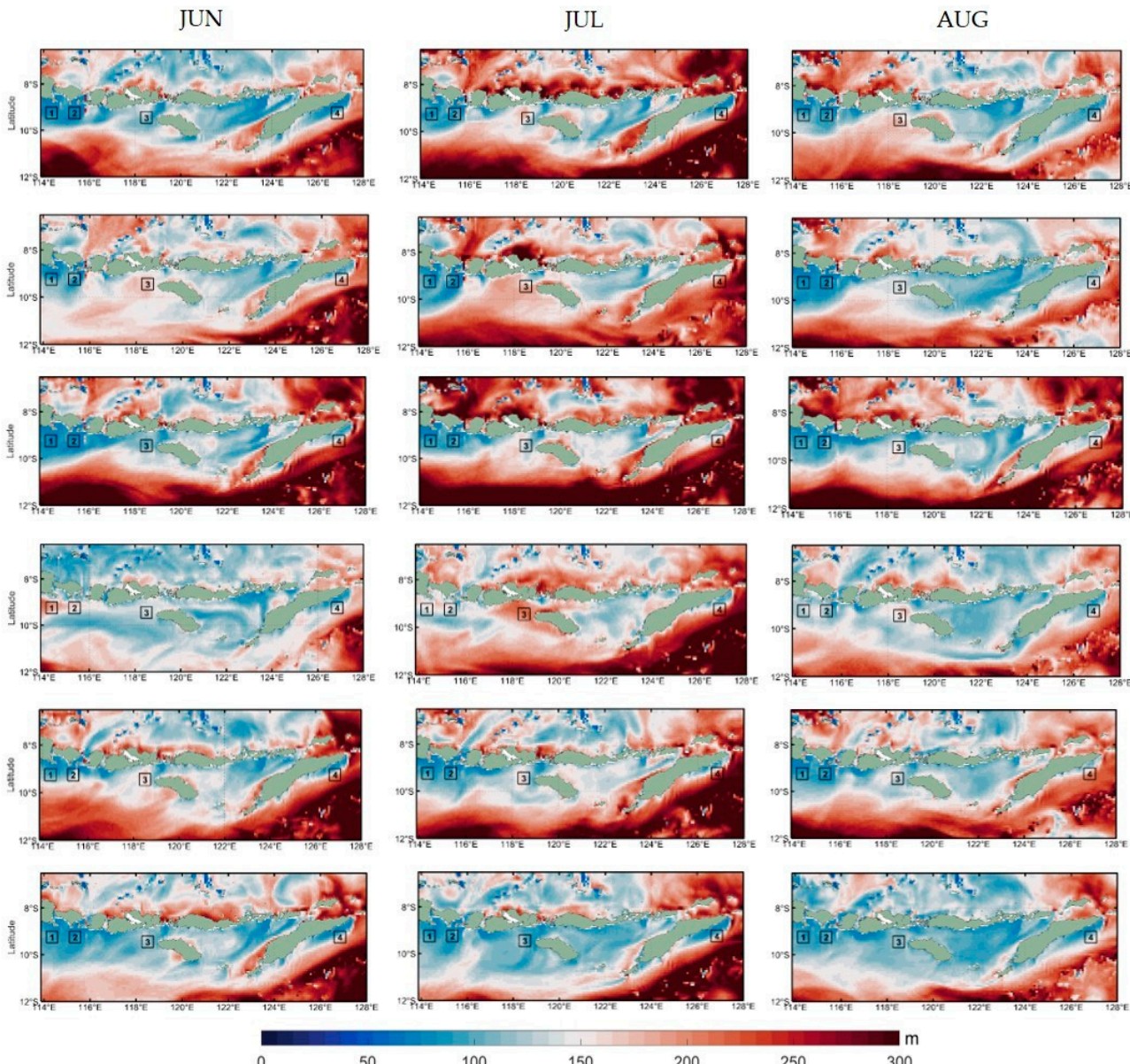

**Figure 19.** The mixed layer depth in the 2007 La Nina, 2009 El Nino, 2008 positive Indian Ocean Dipole (IOD), 2016 negative IOD, 2015 positive IOD and El Nino, and 2010 negative IOD and La Nina, respectively (from top to bottom).

Additionally, the deepest mixed layer occurred in July of the 2016 negative IOD, ranging from 139 to 189 m. These results are consistent with the negative anomaly of chl-a concentration and the positive anomaly of SST in all boxes, corresponding to the negative anomalies of wind stress and EMT. Furthermore, Box 4 provides the deepest mixed layer among all boxes in all events. Based on the climatological data, this area has a lack of upwelling strength, as shown in Figure 8. In addition, the EPV and the meridional EMT intensity are also not as strong as in the other boxes, as shown in Figure 7, which may play a crucial role in defining the mixed layer depth. Deeper mixed layers inhibit phytoplankton blooms on the ocean surface since they do not allow nutrient-rich water to be entrained to the surface, resulting in a decrease in cold and rich-nutrient water along the southern coast of LSI. By comparison, Box 1 yields the shallowest mixed layer depth in all events. This is understandable because, as illustrated in Figure 8, this area shows a strong upwelling intensity during the southeast monsoon season based on the climatology data. In addition, as seen in Figure 7, there is a strong meridional EMT and EPV intensity in Box 1, which

causes a coastal upwelling that lifts cold and rich-nutrient water from a subsurface layer to the ocean surface.

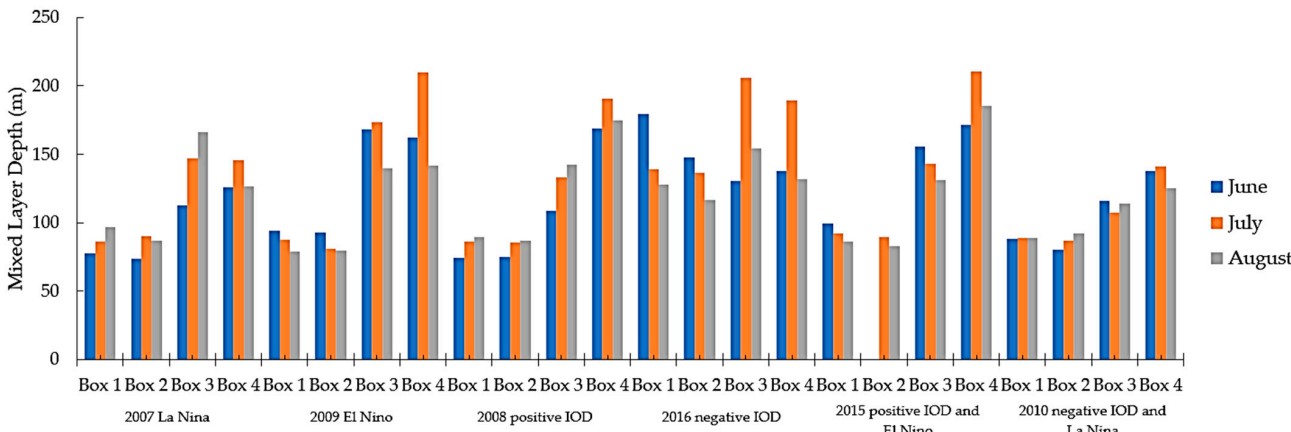

**Figure 20.** The average mixed layer depth in all boxes.

**Table 3.** Correlation between monthly averages of SST, ONI, and DMI in all boxes from 2007 to 2020.

|  | SST Anomalies | | | |
| --- | --- | --- | --- | --- |
|  | **Box 1** | **Box 2** | **Box 3** | **Box 4** |
| ONI | 0.001 | −0.05 | 0.06 | 0.05 |
| DMI | −0.57 | −0.58 | −0.53 | −0.41 |

## 4. Discussion

The southern coast of LSI has long been regarded as an essential upwelling location that feeds the local fisheries. Remotely sensed chl-a can reveal high productivity. Chl-a is a vital biological component since it is linked to productivity, a key variable in managing marine resources, specifically in the fishery industry [62,63]. Enhanced chl-a is an indication of enhanced primary ocean production [64,65]. Earlier investigations have found that seasonal variations in chl-a are caused by monsoon-driven coastal upwelling, with lower and higher values of chl-a observed during non-upwelling and upwelling seasons, respectively [61,66]. The effect of ENSO and IOD on the ocean surface, and the southern coast of LSI, has been studied [11,41]. In our analysis, based on the higher spatial resolution of sea surface wind remote sensing products, we found that wind stress and EMT play an essential role in defining ocean surface conditions along the southern coast of LSI. For instance, we observed positive wind stress and EMT anomalies during the 2007 La Nina event. Additionally, based on previous studies, La Nina induced a negative chl-a concentration anomaly and a positive SST anomaly [11,40,42].

We also determined the relationship between SST and ENSO (represented by ONI) and IOD (represented by DMI), as shown in Table 3, which previous studies failed to do [11,41]. A negative correlation between DMI and the SST anomaly means that the higher the DMI value, the lower the SST anomaly value, and vice versa. We noticed that, in all boxes, DMI negatively correlated with the SST anomaly. Furthermore, the highest correlation value in Box 2 was −0.58. Additionally, low correlation values were observed between ENSO and the SST anomaly in all boxes. These results suggest that the IOD has a greater influence on SST than ENSO in all boxes. This result is consistent with a previous study on adjacent seas (i.e., the southern coast of Java Sea) [43,67], where the IOD was shown to play a greater role in enhancing upwelling along Java's southern coast compared with ENSO. The author stated that the DMI is more closely linked to the south Java upwelling occurrences of 2003, 2006, 2007, 2008, and 2011 than the Nino-3 index.

We observed four different areas under finer-scale maps of sea surface wind, which show different associations between SST and sea surface wind along the southern coast of LSI. Understanding the characteristics of the specific areas studied in our analysis is essential for marine resource management. Box 1, Box 2, and Box 3 showed cold SSTs and high chl-a concentrations during the southeast monsoon season. This may help people living near coastal areas to increase their fish catch as small fish are attracted to the presence of chl-a, and this attracts large fish. Our study may also be useful for the local government or the Ministry of Marine Affairs and Fisheries to discover other prospective fishing grounds in other regions of LSI.

We also analyzed the mixed layer depth during different phases of ENSO and IOD, as shown in Figure 19. Assessing the mixed layer is crucial since it plays a central role in the oceanic food chain. The initial link in this chain is primary phytoplankton production. In our analysis, we noticed that a shallow mixed layer depth induced a high chl-a concentration and a cold SST, as shown during the 2008 positive IOD, 2009 El Nino, and 2015 positive IOD and El Nino. In addition, we also observed a shallow mixed layer depth during 2007 La Nina due to a strong wind stress and EMT. This result indicates that sea surface wind (represented by wind stress and EMT in this case) plays a vital role in defining the mixed layer depth, which correlated with a high chl-a concentration and a low SST.

## 5. Conclusions

The effects of AAM wind on chl-a, SST, and Ekman dynamics variability along the southern coast of LSI and its connection to ENSO and IOD were investigated in this study using satellite data, e.g., SST, chl-a, and wind data, along with reanalysis data and in situ observations. The findings can be summarized as follows.

(1) The maximum chl-a concentration, of more than 1.5 mg m$^{-3}$, occurs in August, whereas the minimum SST, of lower than 25 °C, also occurs in August along the southern coast of LSI. This trend is also followed by a strong wind stress (more than 0.1 Nm$^{-2}$). These results suggest the importance of AAM winds in regulating ocean surface conditions through the EMT mechanism. The Australian monsoon in June–August induces a strong offshore EMT that is favorable to upwelling conditions. It brings cold and rich-nutrient water from a subsurface layer to the ocean surface. As a result, a high chl-a and a low SST occur in June–August.

(2) Throughout the southeast monsoon season, the relationship between the distribution of SST and sea surface wind speed is incongruent. We observed a lower SST in Box 1 than in Box 2, whereas the wind speed in Box 1 is weaker than in Box 2. This discrepancy is due to Ekman dynamics processes, which affect SST variation along the southern coast of LSI. The SST variability in Box 1 is delineated by a negative EPV and a weak offshore EMT. Conversely, Box 2 is delineated by a strong offshore EMT and a positive EPV. A negative EPV enhances EMT-produced upwelling, despite the fact that the coastal upwelling induced by an offshore EMT in Box 1 is weak. The positive EPV-induced downwelling in Box 2, conversely, counteracts robust offshore EMT-induced upwelling. In addition, we observed a consistent relationship between wind speed and SST in Box 3 and Box 4. SST variability in Box 3 is delineated by a negative EPV and a strong offshore EMT, whereas the combination of a weak offshore EMT and a positive EPV determined the SST variability in Box 4.

(3) The offshore EMT dominates in the seasonal fluctuation of EMT, which is favorable for generating coastal upwelling. The offshore EMT appears for eight months, from April to November, when there is a southeasterly wind, and reaches its maximum value (about 4 m$^2$ s$^{-1}$) in August along the southern coast of Bali, Lombok, and Sumbawa. In addition, the onshore EMT appears for four months, from December to March, when there is a northwesterly wind, and reaches its maximum value (about 2 m$^2$s$^{-1}$) in January. Additionally, negative EPV values (upward water motion) are dominant during the southeast monsoon season. In contrast, positive EPV values (downward water motion) are dominant during the northwest monsoon season.

(4)   Regarding interannual variation, La Nina coincides with negative IOD occurrences, reduces offshore EMT intensity, and induces a low chl-a concentration and a warm SST. In contrast, El Nino and positive IOD tend to strengthen offshore EMT, resulting in a high chl-a concentration and a cold SST. Furthermore, the shallowest mixed layer depth occurred during the 2007 La Nina event, and the positive anomaly of chl-a concentration and the negative anomaly of SST correspond to the positive anomalies of wind stress and EMT. A shallow mixed layer depth allows for phytoplankton blooms on the ocean surface because this enables the nutrient-rich water to be entrained to the surface, resulting in cold and rich-nutrient water along the southern coast of LSI. In addition, the IOD has a more significant impact on the variability in SST than ENSO in all four boxes.

**Author Contributions:** Conceptualization, T.-H.L. and F.S.; methodology, F.S. and T.-H.L.; software, F.S.; validation, F.S.; formal analysis and investigation, T.-H.L. and F.S.; data curation, F.S.; writing—original draft preparation, F.S. and T.-H.L.; writing—review and editing, T.-H.L. and F.S.; supervision, project administration and funding acquisition, T.-H.L. All authors have read and agreed to the published version of the manuscript.

**Funding:** This work was financially supported by the Taiwan Ministry of Science and Technology (MOST) Grant MOST 109-2111-M-008-022 and MOST 109-2621-M-008-006.

**Institutional Review Board Statement:** Not applicable.

**Informed Consent Statement:** Not applicable.

**Data Availability Statement:** Data Availability Statement: The data used in this study are open to the public and free to use. Chlorophyll-a and SST data can be obtained from https://giovanni.gsfc.nasa.gov/giovanni/ (accessed on 25 February 2021), WOD2018 data can be obtained from https://www.ncei.noaa.gov/products/world-ocean-database (accessed on 25 February 2021), SSW data can be obtained from https://resources.marine.copernicus.eu/product-dowload/WIND_GLO_WIND_L3_REP_OBSERVATIONS_012_005 (accessed on 25 February 2021), Reanalysis data can be obtained from https://resources.marine.copernicus.eu/product-download/GLOBAL_REANALYSIS_PHY_001_030 (accessed on 25 February 2021).

**Acknowledgments:** The authors deeply appreciate the MODIS chl-a and SST products provided by NASA GESDISC (https://giovanni.gsfc.nasa.gov/giovanni/ (accessed on 25 February 2021), and sea surface wind products from Met-Op ASCAT satellite and the reanalysis data from GLOBAL_REANALYSIS_PHY_001_030 provided by European Union's Earth Observation Programme (Copernicus Programme), and the in-situ observations of the vertical temperature profile from World Ocean Database (WOD) provided by NOAA. The authors are also very grateful to the editor and reviewers for their efforts in processing and reviewing the manuscript of this work.

**Conflicts of Interest:** The authors declare no conflict of interest. The funders had no role in the design of the study; in the collection, analyses, or interpretation of data; in the writing of the manuscript, or in the decision to publish the results.

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
