# Peer review of "Monsoon Effects on Chlorophyll-a, Sea Surface Temperature, and Ekman Dynamics Variability along the Southern Coast of Lesser Sunda Islands and Its Relation to ENSO and IOD Based on Satellite Observations"

_remotesensing, doi:10.3390/rs14071682_

Round 1

Reviewer 1 Report

This is an interesting analysis of upwelling in the region south of the Lesser Sunda Island chain during various extremes associated with ENSO and the IOD. I found the analysis of chl-a and SST anomalies determined from satellite-borne sensors in the context of Ekman upwelling and oceanward/coastward Ekman transport to be particularly interesting. The English of the text was also quite good for which I thank the authors. I do however believe that the manuscript needs some work prior to publication. In particular, I really struggled with precisely what the authors were referring to when describing anomalies in many of the figures, especially Figs. 13-19. My confusion stems from a lack of clarity with their geographic description of the region. It starts with their reference to the Lesser Sandu Island. The use of the singular here is very confusing. I Googled this and it only appears in the plural “Lesser Sandu Islands”. They could refer to the Lesser Sandu island chain but I do not think that the “Lesser Sandu Island” is correct. I know it sounds trivial but this confused me throughout the manuscript. In addition to this, their analysis is of the southern coast of the Lesser Sandu Island, which they appear to define as “the Savu Sea, which Rote Island and Timor bound in the east, Savu Island to the south, Sumba Island to the west.” But they then define 4 boxes as their specific study area. Maybe I paid too much attention to the sentence related to the Savu Sea but that stuck with me throughout. In retrospect I  now see that they were referring to the Savu Sea as a region within the region they are analyzing and that they discuss it in the introduction because of its biological importance. Unfortunately, the geographic description of this area along with the singular (Island) left be thinking that this was the focus area and this is how I read much of the manuscript. I think that they can clear this up quite easily. While they are doing so they might also want to clarify exactly how far offshore their descriptions go. The boxes appear to be about 50 km offshore but many of the anomalies they appear to be describing are closer to shore. 

I should probably reread the manuscript now that it is more clear to me what the Lesser Sunda Islands are but I don’t have the time for this - I’ll see it in the revision. I’m sorry that I was so dense on the first pass through.

I have made a number of editorial suggestions in the attached copy of the manuscript.

Please feel free to reveal my name to the authors: Peter Cornillon

Reviewer 2 Report

I found this to be a generally good paper worthy of publication in Remote Sensing.  I recommend its publication following minor changes.  I would further suggest a careful grammatical review as I may not have caught all grammatical issues.  Overall, the writing is quite clear and easy to read.  The graphics are very well presented.  I especially like the investigation into the oceanic responses to combined climate modes using multiple indicators of upwelling.  Good job.

Minor comments and suggestions for grammatical revisions:

L22 – delete (represented by four boxes) in the abstract

L 25 – grammar.  Should read “coincide” not “coincides”

Abstract – last sentence does not make much sense because the reader has no idea what boxes 1 and 2 are at this point.

L32 – grammar.  Should read “a higher layer”

Figure 1 caption – grammar.  Should read “marked by alphabet letters”

L61 “are” not “is”

L 72 “induces” not “induced”

L93 grammar issue.  Either “Previous studies have used” or “This previous study used”, etc.

jL98 “have not” not “has not”

L 102 would sound better as “The satellite observation and reanalysis products”

L 115 SSC not yet defined

Eq 2a and 2b are confusing…they seem to both have the same U10 range. 

All lat/lon values can probably be represented without the use of seconds (which are always 00 throughout the paper).

Table 2 and L 399-416 – I don’t doubt the correlations, but it is important here to remind the reader of the time series being correlated.  I believe they are monthly averages of box-averaged values for 2007-2020?

L443 “deepens”

Table 3 – Suggest converting to a graph (a bar plot would be appropriate).

L552 – The sentence indicates that variations in chl-a control primary production, but I think the opposite is true (enhanced chl-a is an indication of enhanced primary production).

L554 – “seasonal variations in chl-a are caused by…”

Table 4 – this table is not referenced in the text (maybe mistakenly referenced as table 3 in line 564).

Table 4 caption – again, describe the time series that are used to compute the correlations (computed from monthly averaged values for some year range, for example).
